# Anisotropic flexibility and rigidification in a TPE-based Zr-MOFs with scu topology

Sha-Sha Meng[1,9], Ming Xu[1,9], Hanxi Guan[2,3,9], Cailing Chen [4], Peiyu Cai[5], Bo Dong[1], Wen-Shu Tan[1], Yu-Hao Gu[1], Wen-Qi Tang[1], Lan-Gui Xie [1], Shuai Yuan [6], Yu Han [4,7,8], Xueqian Kong [2] & Zhi-Yuan Gu [1]✉

Tetraphenylethylene (TPE)-based ligands are appealing for constructing metal-organic frameworks (MOFs) with new functions and responsiveness. Here, we report a non-interpenetrated TPE-based scu Zr-MOF with anisotropic flexibility, that is, Zr-TCPE ($H_4$TCPE = 1,1,2,2-tetra(4-carboxylphenyl)ethylene), remaining two anisotropic pockets. The framework flexibility is further anisotropically rigidified by installing linkers individually at specific pockets. By individually installing dicarboxylic acid $L_1$ or $L_2$ at pocket A or B, the framework flexibility along the b-axis or c-axis is rigidified, and the intermolecular or intramolecular motions of organic ligands are restricted, respectively. Synergistically, with dual linker installation, the flexibility is completely rigidified with the restriction of ligand motion, resulting in MOFs with enhanced stability and improved separation ability. Furthermore, in situ observation of the flipping of the phenyl ring and its rigidification process is made by $^2$H solid-state NMR. The anisotropic rigidification of flexibility in scu Zr-MOFs guides the directional control of ligand motion for designing stimuli-responsive emitting or efficient separation materials.

Metal-organic frameworks (MOFs) with adjustable porosity and tunable functionality have attracted considerable attention and exhibited enormous potential in various applications[1–4]. The structures and properties of MOFs could be topologically designed by judiciously selecting organic ligands and metal components[5–7]. Tetraphenylethylene (TPE)-based ligands with aggregation-induced emission (AIE) characteristics have been widely used in various fields, such as organic light-emitting diodes, photodynamic therapy agents, chemo- and bio-sensors[8–11]. The emission of TPE-based ligands is greatly correlated with intramolecular motion and intermolecular stacking states[12,13]. Such unique optical properties of these ligands provide the opportunity for constructing stimuli-responsive MOFs with new functions[14–19].

The $Zr_6$ cluster has been widely used to construct stable MOFs due to its tunable connectivity[20–23]. Up to now, some TPE-based Zr-MOFs with different topologies, such as ftw, csq, and scu, have been reported[24–29]. As to the TPE-based Zr-MOFs with ftw topology[24,28], although the phenyl flipping of organic ligands exists, the organic ligands are firmly and identically immobilized into the framework along a, b, and c directions, and the framework is rigid (Fig. 1). In the csq topology[25], the conformation of organic ligands changes accordingly as the framework shrinks or expands in a one-dimensional (1-D)

¹Jiangsu Key Laboratory of Biofunctional Materials, Jiangsu Collaborative Innovation Center of Biomedical Functional Materials, Jiangsu Key Laboratory of New Power Batteries, College of Chemistry and Materials Science, Nanjing Normal University, Nanjing 210023, China. ²Department of Chemistry, Zhejiang University, Hangzhou 310027, China. ³Institute of Zhejiang University-Quzhou, Quzhou 324100, China. ⁴Advanced Membranes and Porous Materials Center, Physical Sciences and Engineering Division, King Abdullah University of Science and Technology, Thuwal 23955-6900, Saudi Arabia. ⁵Department of Chemistry, Texas A&M University, College Station, TX 77843-3255, USA. ⁶State Key Laboratory of Coordination Chemistry, Key Laboratory of Mesoscopic Chemistry of MOE, School of Chemistry and Chemical Engineering, Nanjing University, Nanjing 210023, China. ⁷Electron Microscopy Center, South China University of Technology, Guangzhou 510640, China. ⁸School of Emergent Soft Matter, South China University of Technology, Guangzhou 510640, China. ⁹These authors contributed equally: Sha-Sha Meng, Ming Xu, Hanxi Guan. ✉e-mail: guzhiyuan@njnu.edu.cn

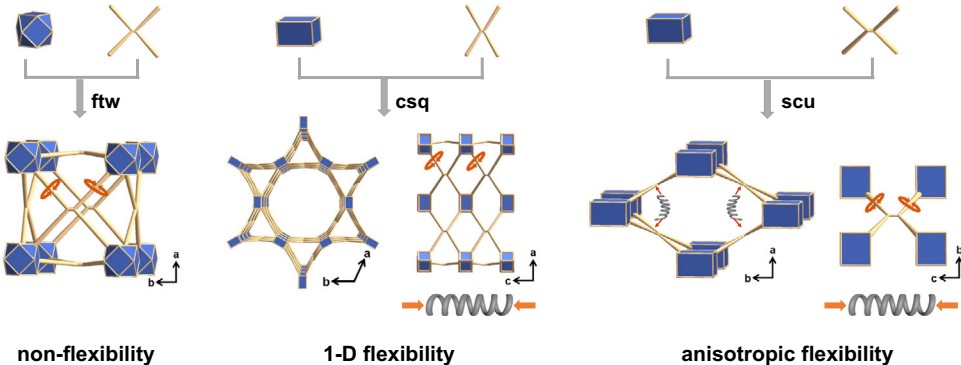

**Fig. 1 | The illustration of TPE-based Zr-MOFs with ftw, csq, and scu topology to show different flexibility, respectively.** The blue polyhedrons represent $Zr_6$ clusters, and the yellow rods represent the organic ligands.

direction along the *c*-axis, while the stacking state of organic ligands on the ab plane changes insignificantly (Fig. 1). Comparatively, the Zr-MOFs with scu topology, possessing one-dimensional channels, have the potential to separately display the intramolecular motion and intermolecular stacking of organic ligands in a three-dimensional (3-D) anisotropic manner. On the one hand, the variation of the framework along the channel direction is closely related to the intramolecular motion of organic ligands. On the other hand, the change of framework in the a or b directions perpendicular to the channel will affect the intermolecular stacking state of ligands (Fig. 1). Thus, constructing the anisotropically flexible TPE-based Zr-MOFs with scu topology is a feasible approach to directionally investigate the intermolecular and intramolecular motions of organic ligands. The scu Zr-MOF (LIFM-114) constructed with ETTC ligand has been synthesized with a 2-fold interpenetrated structure, which made it complex to analyze the different motions of organic ligands[27]. Although, very recently, a non-interpenetrated scu Zr-MOF has been reported, the inter- and intra-molecular motions of organic ligands have yet not been investigated[29].

The exploration of directional intermolecular and intramolecular motion requires the anisotropic control of framework flexibility. Linker installation has been reported as an efficient strategy to rigidify the framework flexibility by inserting secondary linkers into a specific position of pristine MOFs[30–38]. For Zr-MOFs with scu topology, there are two types of vacant coordination pockets available for linker installation. Thus, it is possible to control the anisotropic flexibility of scu MOFs by installing different second linkers at different positions. However, previous studies only reported the enhancement of the framework stability through linker installation by connecting the two scu interpenetrated frameworks[26]. The directional control of inter-molecular and intramolecular motion of TPE-based ligands in the non-interpenetrated flexible MOFs has not yet been studied.

Here, we reported an anisotropically flexible non-interpenetrated TPE-based Zr-MOF with scu topology. Through deliberately installing linkers along or perpendicular to the channel direction of scu MOFs, anisotropic framework flexibility was rigidified, and the intermolecular or intramolecular changes of TPE-based ligands were studied separately. In specific, the TPE-based ligand, 1,1,2,2-tetra(4-carboxylphenyl) ethylene ($H_4TCPE$), was selected to construct the MOF, Zr-TCPE (Fig. 2a). The simulation of HOMO-LUMO plots demonstrated the molecular orbitals upon the conformational changes of Zr-TCPE. According to size matching, different linear linkers, naphthalenedi-carboxylic acid ($L_1$) and fumaric acid ($L_2$), were selected to install into Zr-TCPE at vacant coordination pocket A and pocket B, respectively, further to rigidify the anisotropic flexibility of Zr-TCPE. The $^2H$ solid-state NMR spectra (SSNMR) showed the in situ flipping of phenyl rings of TCPE ligands in Zr-TCPE and its rigidification in Zr-TCPE-$L_1$. Besides, the fluorescence spectra of Zr-TCPE before and after linker installation illustrated the role of intermolecular and intramolecular motions. On

the one hand, the decrease of the spacing of pocket A resulted in the close intermolecular stacking of TCPE ligands. The installation of $L_1$ at pocket A rigidified the flexibility of Zr-TCPE on the ab plane. On the other hand, the linker installation at pocket B rigidified the flexibility of Zr-TCPE along the *c*-axis and restricted the intramolecular motion of TCPE ligands, resulting in enhanced chemical stability and fluorescent stability. Finally, the rigidified TPE-based Zr-MOFs with scu topology showed good separation efficiency for isomers. The elucidation of the inter-/intramolecular motions of TPE-based ligands and the flexibility rigidification of Zr-MOFs with scu topology paves the road to design-ing highly efficient emitter and separator.

## Results and discussion
### Characterizations of Zr-TCPE
Zr-TCPE was solvothermally synthesized by reacting $ZrCl_4$ and $H_4TCPE$ in N, N'-dimethylformamide (DMF) solution at 120 °C with acetic acid (AA) and water as modulators (Fig. 2)[39,40]. Due to the flexibility of the organic ligand and the fast nucleation between the Zr and this organic ligand, large-sized single crystals of Zr-TCPE were hard to obtain, although extensive attempts had been conducted. Thus, the powder X-ray diffraction (PXRD) Rietveld refinement was utilized to reveal the structure of this material (Supplementary Fig. 1)[29]. The refinement results indicated that Zr-TCPE crystallizes in orthorhombic crystal system with *Cmmm* space group. The diffraction peaks at 5.85°, 7.29°, 9.35°, 10.13°, 11.72°, and 12.49° represented the (020), (001), (021), (130), (040), and (130) planes in Zr-TCPE, respectively (Supplementary Table 3). Detailed crystallographic parameters, atomic positions, and diffraction parameters were given in Supplementary Information (Supplementary Tables 1–3). The crystallographic parameters were slightly different from the reported scu TPE-based MOFs[29]. Each $TCPE^{4-}$ ligand was connected to four $Zr_6$ clusters, while each $Zr_6$ cluster was coordinated with eight $TCPE^{4-}$ ligands, generating a non-interpenetrated scu structure. The unsaturated metal sites were occupied by terminal $H_2O$ or $OH^-$ groups and acetate. The Zr-TCPE exhibited scu topology with a fence shape, generating rhombic chan-nels of 9.3 × 23.0 Å along the [001] direction and 5.2 × 11.7 Å along the [100] direction (Supplementary Figs. 2–4). There were two vacant coordination pockets of different sizes in Zr-TCPE, namely, pocket A and pocket B (Fig. 2a).

The non-interpenetrated structure of Zr-TCPE was further con-firmed by low-dose high-resolution transmission electron microscopy (HRTEM) imaging (Fig. 2)[41–43]. The low electron dose (only a few elec-trons per square angstrom) of HRTEM avoids the structural damage of Zr-TCPE under electron beams. The black dots in the HRTEM images represented the $Zr_6$ clusters. The fast Fourier transform (FFT) pattern of the marked area in the HRTEM image was acquired. The calculated d-spacing of (020), (001), and (021) was 1.53 nm, 1.26 nm, and 0.95 nm, respectively, which was consistent with the d-spacing from PXRD

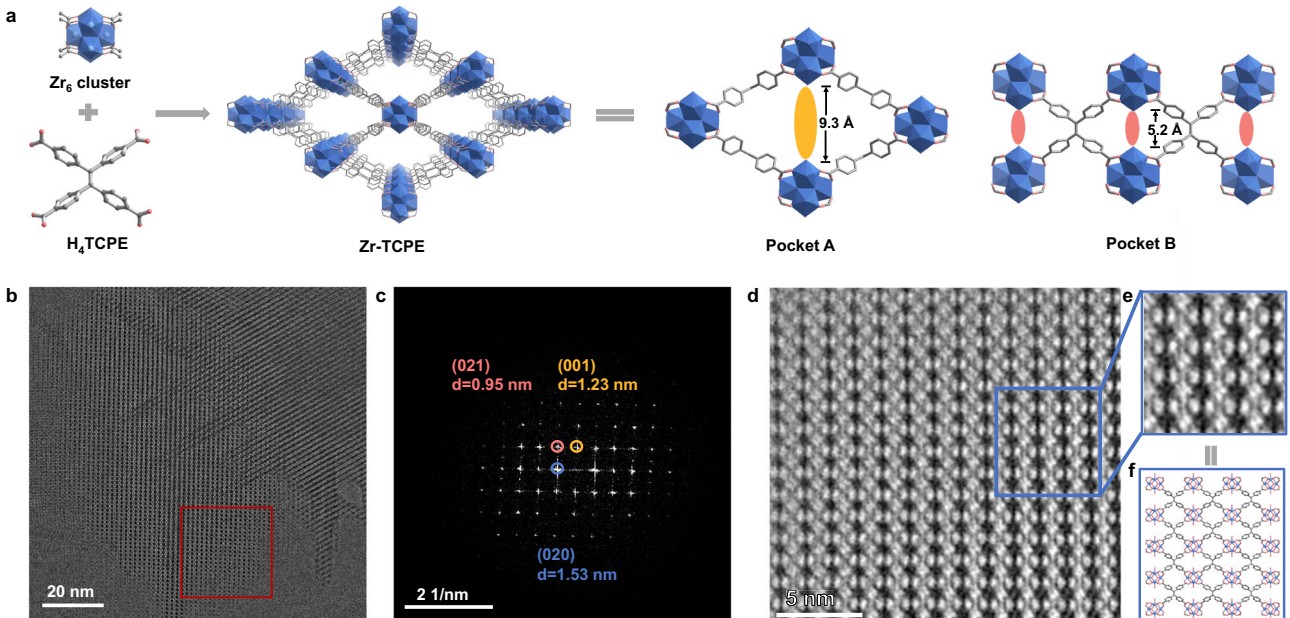

**Fig. 2 | Schematic illustration and characterization of Zr-TCPE. a** The construction of Zr-TCPE from $Zr_6$ cluster and $H_4$TCPE ligand. The illustration of pocket A along [001] direction and pocket B along [100] direction in Zr-TCPE. **b** HRTEM image of Zr-TCPE acquired along the [100] direction. **c** FFT pattern of the marked region in the HRTEM image. The d-spacing of (020), (021), and (001) were calculated at 1.53 nm, 0.95 nm, and 1.23 nm, respectively. **d** ABSF-filtered CTF-corrected image of the marked region in the HRTEM image. The black dots in the HRTEM images represented the $Zr_6$ clusters. **e** The locally enlarged image. The TCPE ligand was even observed in the center of four adjacent $Zr_6$ clusters. **f** Simulated structure of Zr-TCPE along [100] direction.

refinement (Fig. 2c and Supplementary Table 3). For better interpretation, the raw image was processed by correcting the effect of the contrast transfer function (CTF) of the objective lens (Supplementary Fig. 5). Furthermore, the simulated electron diffraction (ED) pattern along the [100] direction was consistent with the selected area electron diffraction (SAED) pattern of the HRTEM image (Supplementary Fig. 6). The average background subtraction filter (ABSF)-filtered CTF-corrected image in Fig. 2d matched well with the simulated structure of Zr-TCPE along the [100] direction zone axis (Supplementary Fig. 3). The distances between two adjacent $Zr_6$ clusters were measured from the linear profiling along the $b$- and $c$-axis as 1.53 nm and 1.23 nm, respectively (Supplementary Fig. 7), which were consistent with the simulated structure of Zr-TCPE (Supplementary Fig. 3). Besides, the distance of $Zr_6$ clusters along the $c$-axis was comparable to the $c$-axis cell parameter in non-interpenetrated Zr-TCPE. Thus, not interpenetrated but non-interpenetrated structure was assigned to Zr-TCPE. We also obtained diffraction information of (020) and (130) planes from the FFT images (Supplementary Fig. 8). All the above results verified the non-interpenetrated structure of Zr-TCPE.

The high-angle annular dark-field (HAADF) images of Zr-TCPE were also collected to further character the diffraction information (Supplementary Figs. 9 and 10). The denoised image and FFT pattern were also acquired. The white dots in these images represented the $Zr_6$ clusters. The calculated d-spacing of the (020) plane was 1.54 nm, which was comparable to d = 1.51 nm from XRD refinement. Besides, as shown in Supplementary Fig. 10, the distance between two adjacent $Zr_6$ clusters was calculated from the linear profiling in two directions. Due to the limitation of resolution, the distance between two adjacent $Zr_6$ clusters along the $b$-axis was hard to observe directly because of the interference from $Zr_6$ clusters along the $a$-axis. Thus, the denoised images were acquired by an inverse-FFT process after applying a periodic mask to the FFT pattern for measuring the distance of $Zr_6$ clusters. The measured distance of adjacent $Zr_6$ clusters was around 1.79 nm and 3.04 nm (Supplementary Fig. 10). These values were consistent with the simulated value along the $a$-axis and $b$-axis, respectively, in a non-interpenetrated structure but differed from the

distance of adjacent $Zr_6$ clusters in a double-interpenetrated structure (0.89 nm and 1.51 nm, see Supplementary Figs. 3 and 4). The results further proved that the Zr-TCPE possessed a non-interpenetrated structure.

The synthesized Zr-TCPE exhibited a wool ball shape with an average size of about 100 nm (Supplementary Fig. 11). The ¹H NMR result of the digested Zr-TCPE confirmed the existence of TCPE ligands, DMF solvents, and acetic acid in the framework (Supplementary Fig. 12). To accurately characterize the porosity of Zr-TCPE, both the $N_2$ and Ar adsorption-desorption isotherms were performed at 77 K and 87 K, respectively. As shown in Supplementary Fig. 13, both the isotherms presented fully reversible type-I behavior, indicating the microporous characteristic of Zr-TCPE. The Brunauer-Emmett-Teller (BET) surface area, pore size distribution, and total pore volume obtained from the Ar adsorption isotherms are consistent with those from $N_2$ adsorption isotherms (Supplementary Table 4). The main pore size of Zr-TCPE calculated by the DFT method from Ar and $N_2$ sorption isotherms was 8.7 Å and 9.4 Å, respectively. The calculated pore size was comparable to the simulated size of pocket A in Zr-TCPE (9.3 Å), indicating the non-interpenetrated feature of Zr-TCPE.

**Anisotropic flexibility of Zr-TCPE**

The conformation changes of TPE-based ligands in response to external stimuli bring flexibility to MOFs. This flexibility could be reflected in the variation of emission properties of MOFs, which is usually correlated with AIE characteristics of TPE-based ligands[27,44,45]. Thus, we tested the emission properties of Zr-TCPE under the external stimuli of temperature. Under different heating temperatures of 200, 220, and 250 °C, the fluorescence emission of Zr-TCPE red-shifted from 460 nm to 470 nm and 517 nm, respectively (Fig. 3d and Supplementary Fig. 14). After heating Zr-TCPE under 250 °C (Zr-TCPE-H), the color of Zr-TCPE changed from white to yellow (Supplementary Fig. 15a). Meanwhile, the photoluminescent color changed from blue to light green under the 365 nm excitation (Supplementary Fig. 15b). It has been reported that the color change and the fluorescence red-shift of TPE-based MOFs resulted from the intramolecular motion and close stacking of TPE

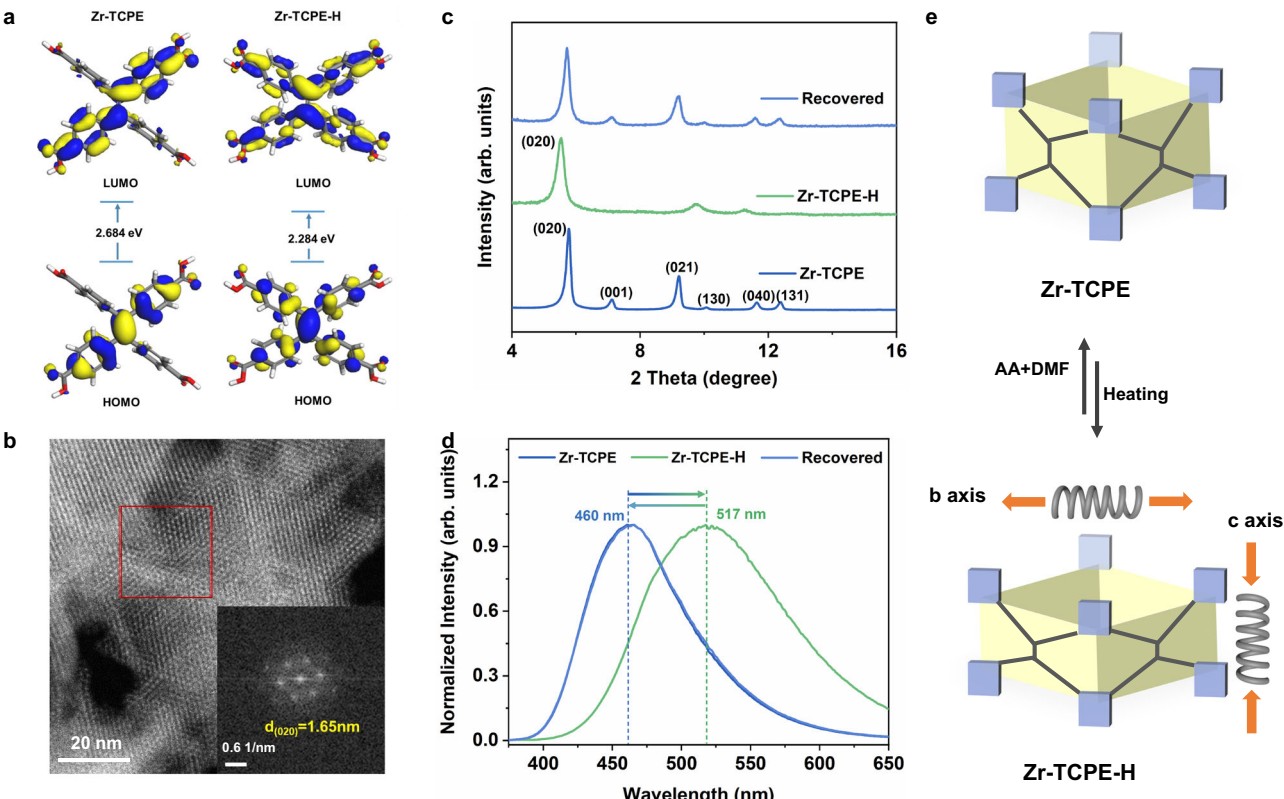

**Fig. 3 | Characterization and structural illustration of anisotropic flexibility in Zr-TCPE. a** The calculated HOMO and LUMO plots of $H_4$TCPE in Zr-TCPE conformation and Zr-TCPE-H conformation. **b** The HAADF images of Zr-TCPE. The white dots represented the $Zr_6$ clusters. The insert is FFT pattern. The d-spacing of the (020) plane was measured at around 1.65 nm. **c** The PXRD patterns of Zr-TCPE, Zr-TCPE-H, and restored Zr-TCPE. **d** The fluorescence emission spectra of Zr-TCPE, Zr-TCPE-H, and restored Zr-TCPE dispersed in methanol solution ($\lambda_{ex}$ = 340 nm). **e** The simplified illustration of anisotropic flexibility and reversible phase transformation of Zr-TCPE.

ligands[27,45]. In the case of Zr-TCPE, the motion and stacking changes were caused by the loss of solvent molecules and coordinated modulators in vacant coordination pockets, which was evidenced by the $^1$H-NMR of digested Zr-TCPE-H (Supplementary Fig. 16). The intramolecular motion of TCPE ligands in Zr-TCPE was associated with the framework change along the $c$-axis, indicating the framework flexibility along the $c$-axis. The close stacking of TCPE ligands in Zr-TCPE corresponded to the framework change on the ab plane (Supplementary Fig. 3). To demonstrate the conformational change and electric states of organic ligands, DFT calculations were performed on $H_4$TCPE ligands in Zr-TCPE and Zr-TCPE-H, respectively. The ligand exhibited a more twisted conformation in Zr-TCPE-H. Besides, the energy gap between the HOMO and LUMO was 2.684 eV for Zr-TCPE, while it was 2.284 eV for Zr-TCPE-H (Fig. 3a). The decreased energy gap resulted in lower emission energy for Zr-TCPE-H, which was consistent with the fluorescence results.

To further investigate the structural change, the PXRD patterns, HAADF images, Raman spectra, transmission electron microscopy (TEM), and scanning electron microscopy (SEM) images of Zr-TCPE-H were collected. The PXRD patterns showed that the first diffraction peaks of the heated Zr-TCPE shifted to a smaller degree (Fig. 3c), indicating the increase of interplanar spacing of the (020) plane. The HAADF images of Zr-TCPE-H also proved that the d value of the (020) plane increased from 1.54 nm to 1.65 nm (Fig. 3b and Supplementary Fig. 17). Besides, the measured distance of adjacent $Zr_6$ clusters along the $b$-axis had also increased from 3.04 nm to 3.25 nm (Supplementary Fig. 18). All the results demonstrated the framework change along the $b$-axis after heating treatment, indicating the framework flexibility along the $b$-axis. It was worth noting that the measured distance of adjacent $Zr_6$ clusters along the $a$-axis displayed no significant changes possibly due to the technical resolution limitation (Supplementary

Fig. 18). In addition, the main pore size of Zr-TCPE-H was 6.8 Å, smaller than that of Zr-TCPE (Supplementary Fig. 19). The reduced pore size proved the shrinkage along the $a$-axis, indicating the framework flexibility along the $a$-axis. The above experiments and characterizations demonstrated the anisotropic flexibility of Zr-TCPE (Fig. 3e). These phenomena illustrated the transformation from Zr-TCPE to Zr-TCPE-H. The SEM and TEM images of Zr-TCPE-H still displayed the original morphology, which excluded the possibility of framework collapse (Supplementary Fig. 20). The Raman peak at 412 $cm^{-1}$ was assigned to the stretching modes of Zr-O (carboxylate), and the peak at 1125 $cm^{-1}$ was related to the stretching vibration of the C-C bond formed between ethylene and phenyl ring (Supplementary Fig. 21). Another obvious Raman peak at 1607 $cm^{-1}$ was assigned to the stretching vibration of the C = C band from phenyl rings and ethylene core[45–47]. Compared with Zr-TCPE, neither Raman peaks disappeared nor new Raman peaks appeared in Zr-TCPE-H, indicating unchanged chemical composition in Zr-TCPE after heating[45,46]. To explore whether this transformation is reversible, another external stimulus is needed to restore the original structure. Then, these heated materials were treated in DMF solution with acetic acid under different temperatures for 12 h (see Supplementary Information for details). The use of temperature is important to restore the structure. At 100 °C or 120 °C, the PXRD patterns and fluorescence emission were completely restored with the addition of acetic acid (Fig. 3d and Supplementary Figs. 22–24). These results proved the reversible transformation of Zr-TCPE, further proving the anisotropic flexibility of Zr-TCPE.

### Anisotropic rigidification of flexibility in Zr-TCPE

To directionally control the anisotropic flexibility of Zr-TCPE, we employed the linker installation strategy. In the scu topology of Zr-

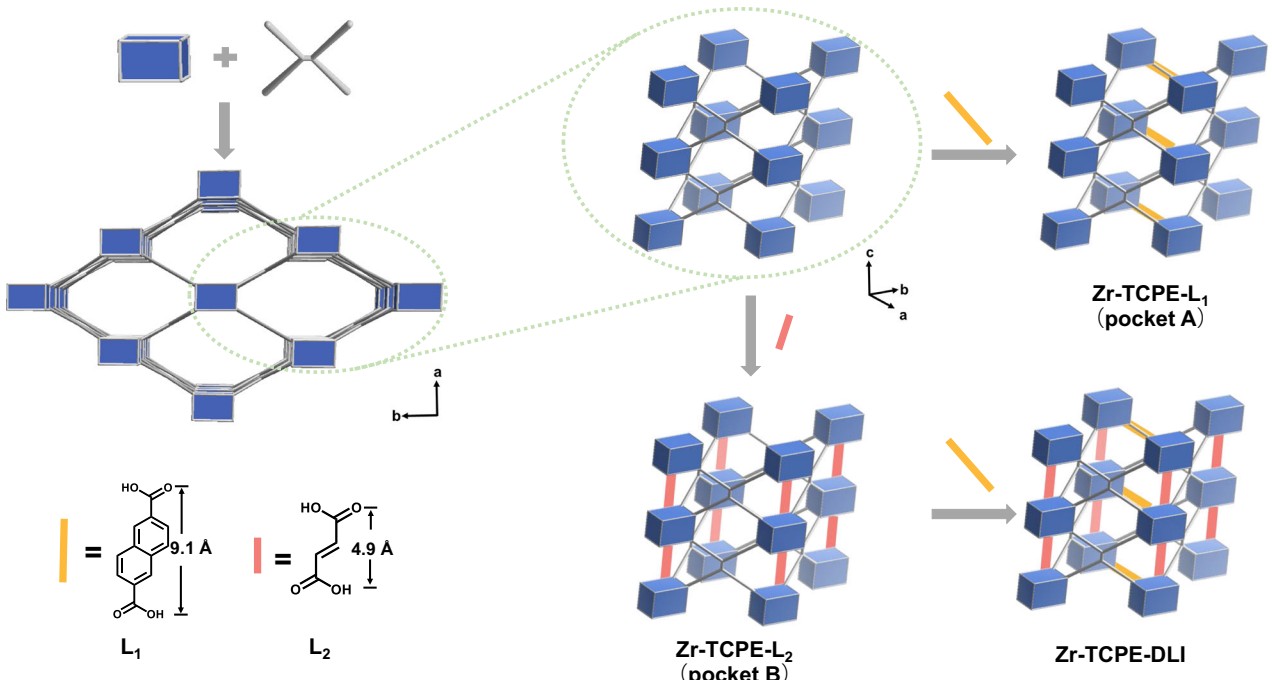

**Fig. 4 | Schematic illustration of linker installation in Zr-TCPE at pocket A and pocket B.** The linkers $L_1$ and $L_2$ were installed at pocket A and B, respectively. DLI represents dual linker installation. The longer yellow rod represents $L_1$ with a length of 9.1 Å, and the shorter red rod represents $L_2$ with a length of 4.9 Å. Zr, blue; O, red; C, light gray; hydrogen atoms omitted for clarity.

TCPE, each $Zr_6$ cluster was coordinated with eight TCPE$^{4-}$ ligands, leaving two types of vacant coordination pockets, packet A along the $c$-axis and pocket B along the $a$-axis (Fig. 2a). The space of pocket A is about 9.3 Å, which is almost equal to the length of naphthalenedicarboxylic acid ($L_1$, 9.1 Å). Meanwhile, the space of pocket B (5.2 Å) is similar to the length of fumaric acid ($L_2$, 4.9 Å). Thus, by post-synthetic modification, $L_1$ and $L_2$ could coordinate on the unsaturated metal sites in pocket A and pocket B, respectively (Fig. 4). This strategy allowed the precise control of anisotropic flexibility in Zr-TCPE and separately investigated the TPE motions along different directions. Through soaking fresh Zr-TCPE in DMF solution of the excessive amounts of installed linkers at 75 °C for 24 h under stirring, Zr-TCPE-$L_1$ and Zr-TCPE-$L_2$ were synthesized. Meanwhile, with the dual linker installation (DLI), the Zr-TCPE-DLI was also obtained, in which the installation order is installing $L_2$ first and then $L_1$.

To quantitatively analyze the installed linkers, the $^1$H NMR spectra of the digested materials after installation were measured, proving the successful introduction of the two linkers (Supplementary Figs. 25–34). The ratios of the installed linkers to the original ligands in the Zr-TCPE series were calculated (Supplementary Table 5). Ideally, if the vacant coordination pockets in Zr-TCPE are fully occupied with installed linkers, the ligand ratio of $L_1$: $L_2$: TCPE will be 0.5: 0.5: 1. In Zr-TCPE-$L_1$, the ratio of $L_1$ to TCPE was 0.5:1, which was consisted with the theoretical value. Because of the size mismatching between the space of pocket B and the length of $L_1$, the possibility of $L_1$ coordinated on pocket B was ruled out. Thus, pocket A was fully occupied by $L_1$ in Zr-TCPE-$L_1$. The ratio of $L_2$ to TCPE was 0.4:1, which was smaller than the theoretical ratio. It resulted from the steric hindrance of pocket B in the original MOF that prevented full coordination at pocket B. Meanwhile, $L_2$ was small enough to coordinate at pocket A as the dangling linker. Thus, $L_2$ was partially coordinated on both pocket A and B. The ligand ratio of $L_1$: $L_2$: TCPE in Zr-TCPE-DLI was 0.5: 0.1: 1. Considering that $L_1$ could only coordinate on pocket A, while $L_2$ possessed the ability to coordinate on pocket A and B, the $^1$H NMR results of the sequential experiments ($L_2$-$L_1$) illustrated that the $L_2$ first dangled at pocket A was then replaced by $L_1$, leading to the full coordination of $L_1$

at pocket A. We also reversed the insertion sequence ($L_1$-$L_2$), and the ligand ratio of $L_1$: $L_2$: TCPE was calculated as 0.3: 0.3: 1 (see Supplementary Information for details). Besides, the $^1$H NMR of the above MOFs after thermal treatment under 250 °C for 6 h and digestion were measured. For all MOFs after thermal treatment, the ratios of additional linkers to TCPE almost remained the same, exhibiting that the installed linkers are thermally stable under 250 °C (Supplementary Table 5).

Further, the PXRD patterns, SEM, and TEM images of Zr-TCPE-$L_1$, Zr-TCPE-$L_2$, and Zr-TCPE-DLI indicated that the linker installation process did not change the structure and morphology of the original MOF (Supplementary Figs. 35 and 36). The PXRD patterns of Zr-TCPE-DLI immersing in aqueous solutions with different pH values exhibited no obvious difference. Besides, the crystallinity of Zr-TCPE-DLI was well maintained after 15 months of storage, which further indicated the enhanced chemical stability (Supplementary Fig. 37). To characterize the porosity parameters of Zr-TCPE after linker installation, $N_2$ adsorption-desorption isotherms were performed at 77 K. As shown in Supplementary Fig. 38, all the isotherms exhibited reversible type-I behavior, indicating the existence of micropores in MOFs. The BET surface area of Zr-TCPE-$L_1$ and Zr-TCPE-$L_2$ were 670 m$^2$/g and 894 m$^2$/g, respectively, which were smaller than that of Zr-TCPE. As shown in Supplementary Fig. 38b, the main size of pores in Zr-TCPE-$L_1$ was 6.4 Å, which corresponded to the partitioning of the rhombic pores (9.4 Å) by $L_1$ at pocket A. Due to technical limitations, the pore partition phenomenon of the other rhombic pores (~5.2 Å, calculated from the simulated structure) by $L_2$ at pocket B could not be observed from the $N_2$ adsorption-desorption results. Nevertheless, the major pore size of Zr-TCPE-$L_2$ (8.0 Å) was slightly smaller than that of Zr-TCPE (Supplementary Fig. 38b). It was because $L_2$ possessed the ability to coordinate on the metal sites at pocket A, $L_2$ could partly dangle on pocket A and reduce the pore size.

The successful installation of linkers at different pockets along different axis directions in Zr-TCPE provides the potential to anisotropically rigidify the framework flexibility and separately investigate the TPE intermolecular and intramolecular motions. The fluorescence

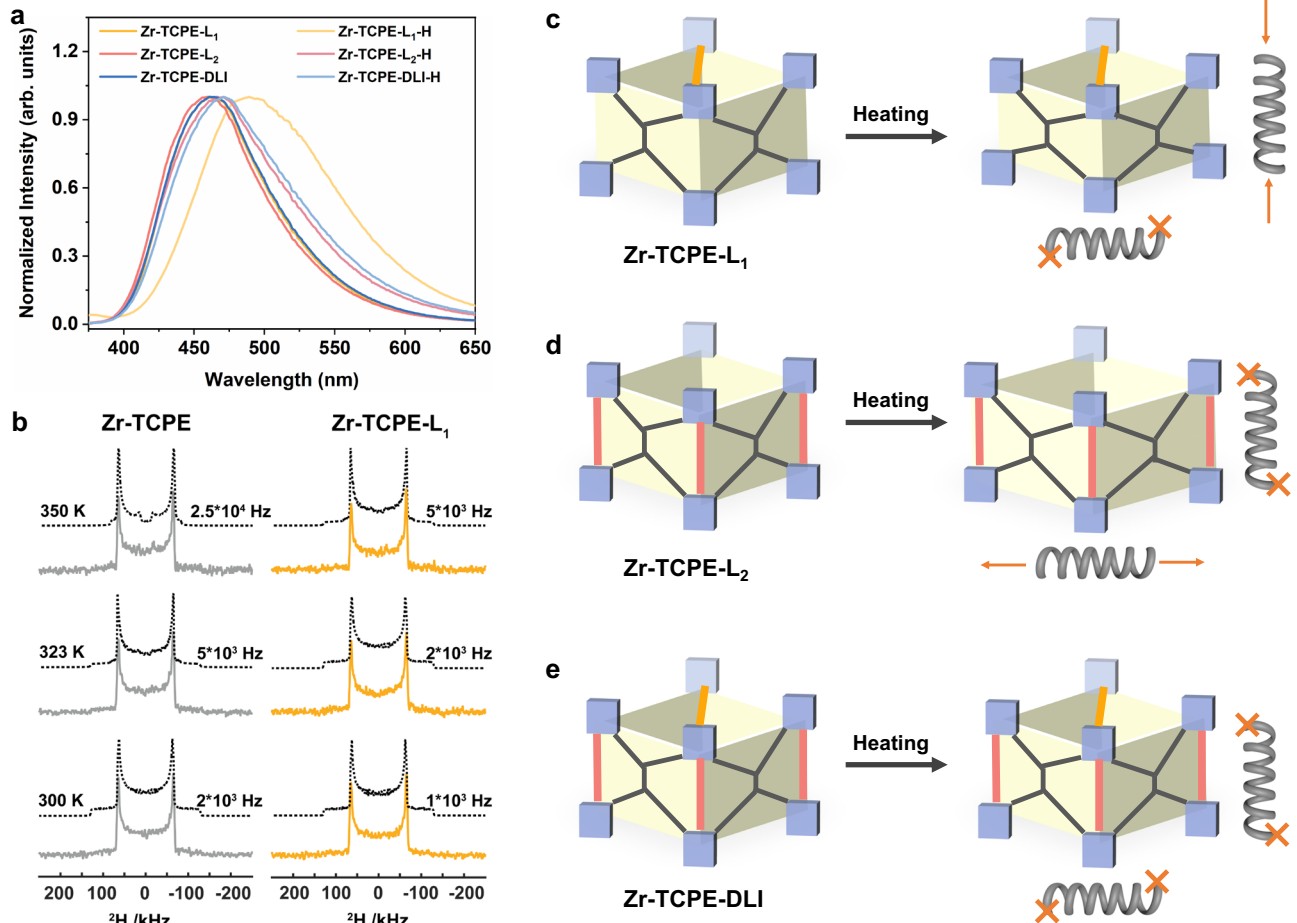

**Fig. 5 | Characterization and structural illustration of anisotropic rigidification in Zr-TCPE by linker installation. a** The fluorescence emission spectra of Zr-TCPE-$L_1$, Zr-TCPE-$L_2$, and Zr-TCPE-DLI before and after heating at 250 °C for 6 h in methanol solution ($\lambda_{ex}$ = 340 nm). **b** Experimental and simulated $^2$H SSNMR spectra of Zr-TCPE (left) and Zr-TCPE-$L_1$ (right) at 300 K, 323 K, and 350 K, respectively. The solid lines represent the experimental results, and the dotted lines represent the simulated results. The illustration of rigidifying the anisotropic flexibility **c** along the *b*-axis in Zr-TCPE-$L_1$, **d** along the *c*-axis in Zr-TCPE-$L_2$, and **e** along both *b*-axis and *c*-axis in Zr-TCPE-DLI.

behaviors of Zr-TCPE-$L_1$, Zr-TCPE-$L_2$, and Zr-TCPE-DLI before and after heating were recorded. Note that these materials were heated at 250 °C for 6 h before characterization, namely Zr-TCPE-$L_1$-H, Zr-TCPE-$L_2$-H, and Zr-TCPE-DLI-H, respectively. As shown in Fig. 5a, the fluorescence emission of Zr-TCPE-$L_2$-H was similar to the original Zr-TCPE-$L_2$. In contrast, the emission of Zr-TCPE-$L_1$-H red-shifted apparently from 460 nm to 490 nm compared with the original Zr-TCPE-$L_1$. The different fluorescence behavior of Zr-TCPE-$L_1$ and Zr-TCPE-$L_2$ was attributed to the different motion states of TCPE ligands in Zr-TCPE. As reported, the fluorescence changes of TPE-based molecules are more dependent on intramolecular motion than intermolecular motion[27,45]. The installation of $L_1$ at pocket A rigidified the flexibility and restricted the intermolecular stacking between adjacent organic ligands on the ab plane. The intramolecular motion of ligands and the flexibility along the *c*-axis still existed in Zr-TCPE-$L_1$ due to the vacant pocket B (Fig. 5c), leading to red-shift fluorescence of Zr-TCPE-$L_1$ under heating. On the other hand, the installation of $L_2$ at pocket B was more likely to rigidify the flexibility along the *c*-axis and restrict the intramolecular motion of organic ligands due to the steric hindrance (Fig. 5d). The intermolecular motion of ligands and the flexibility on the ab plane in Zr-TCPE-$L_2$ had little influence on fluorescence emission. As a result, the fluorescence behavior of Zr-TCPE-$L_2$ exhibited no obvious change after the heating process. Besides, with dual linker installation, the anisotropic flexibility of Zr-TCPE-DLI was completely rigidified (Fig. 5e), which was confirmed by the unchanged fluorescence of Zr-TCPE-DLI. These results indicated that the intermolecular and intramolecular

motion of TPE-based ligands can be investigated with the anisotropic rigidification of framework flexibility through linker installation.

To in situ study the intramolecular dynamics, such as the flipping of phenyl ring motion of organic ligands in Zr-TCPE before and after linker installation, we synthesized deuterated $H_4$TCPE-$d_{16}$ ligands to construct Zr-TCPE and Zr-TCPE-$L_1$ and implemented $^2$H solid-state NMR ($^2$H SSNMR) spectroscopy[48,49]. The detailed synthesis process and characterization of deuterated $H_4$TCPE-$d_{16}$ were given in Supplementary Information (Supplementary Figs. 39–47). The PXRD patterns, TEM images, and $^1$H NMR spectra confirmed the successful synthesis of Zr-TCPE, Zr-TCPE-$L_1$, and Zr-TCPE-$L_2$ constructed with $H_4$TCPE-$d_{16}$ (Supplementary Figs. 48–51). Variable-temperature $^2$H SSNMR spectra of Zr-TCPE and Zr-TCPE-$L_1$ were recorded between 300 K and 350 K. As shown in Fig. 5b, the $^2$H SSNMR spectra of Zr-TCPE and Zr-TCPE-$L_1$ exhibited typical Pake patterns with the splitting of 129 kHz at 300 K, suggesting that the phenyl rings in both samples were almost static. Zr-TCPE at 350 K resulted in a new set of symmetric peaks, while Zr-TCPE-$L_1$ showed almost the same Pake patterns at all temperature ranges. To simulate the $^2$H SSNMR spectra, a phenyl flipping model was carried out using the EXPRESS package running in MATLAB software. The simulated results displayed the flipping rates of phenyl rings in Zr-TCPE were $1 \times 10^3$ Hz, $5 \times 10^3$ Hz, and $2.5 \times 10^4$ Hz at 300 K, 323 K, and 350 K, respectively. With the temperature increasing, the phenyl rings in Zr-TCPE underwent a faster π-flipping motion, as the typical separating pattern of phenyl flipping (33 kHz) in Zr-TCPE was stronger than that in Zr-TCPE-$L_1$. The slower flipping of phenyl rings in Zr-TCPE-

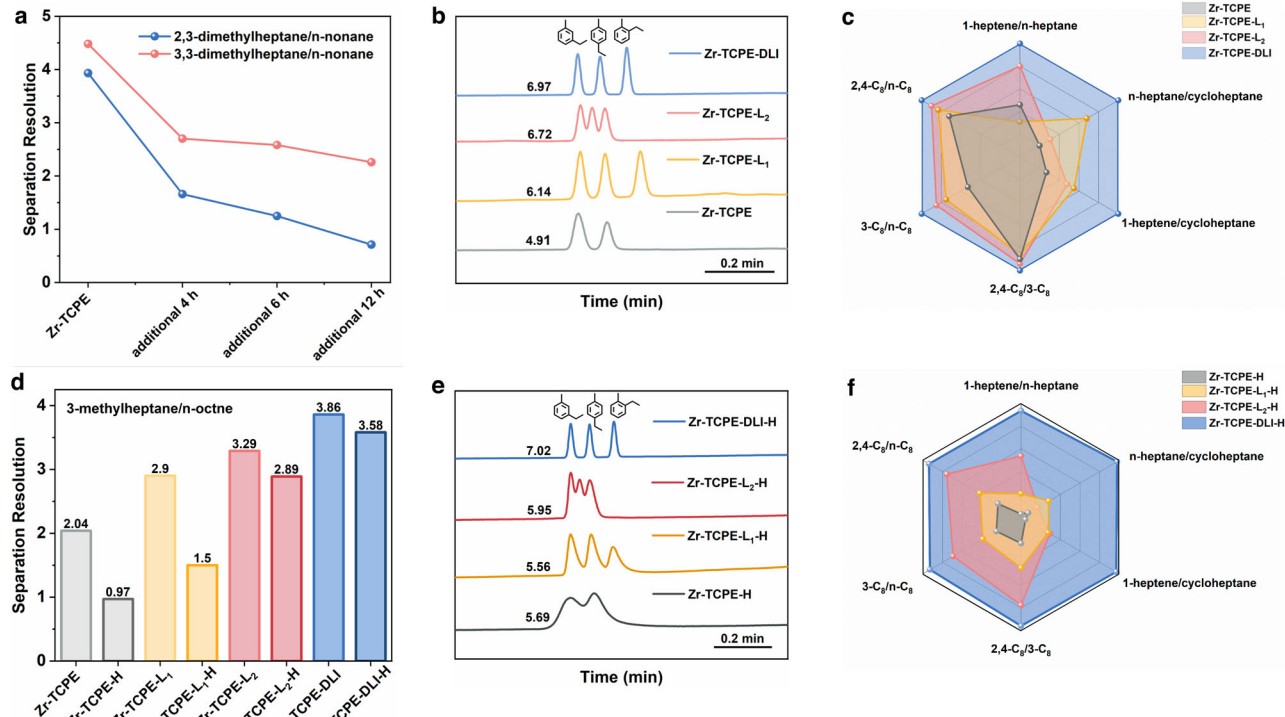

**Fig. 6 | Separation performance. a** Separation resolution of nonane isomers on Zr-TCPE coated column with additional heating at 250 °C for 4, 6, and 12 h. **b** Gas chromatograms of Zr-TCPE, Zr-TCPE-L₁, Zr-TCPE-L₂, and Zr-TCPE-DLI coated columns for the separation of ethyltoluene isomers. The numbers on the left represent the elution time of the first peak. **c** Separation resolution of octane isomers and the mixture of 1-heptene, n-heptane, and cycloheptane on Zr-TCPE, Zr-TCPE-L₁, Zr-TCPE-L₂, and Zr-TCPE-DLI coated columns. 2,4-C₈, 3-C₈, and n-C₈ represent 2,4-dimethylhexane, 3-methylheptane, and n-octane, respectively. The scale was normalized to Zr-TCPE-DLI. **d** Separation resolution for 3-methylheptane and n-octane isomers on Zr-TCPE, Zr-TCPE-H, Zr-TCPE-L₁, Zr-TCPE-L₁-H, Zr-TCPE-L₂, Zr-TCPE-L₂-H, Zr-TCPE-DLI, and Zr-TCPE-DLI-H coated columns. **e** Gas chromatograms of Zr-TCPE-H, Zr-TCPE-L₁-H, Zr-TCPE-L₂-H, and Zr-TCPE-DLI-H coated columns for the separation of ethyltoluene isomers. **f** Separation resolution of octane isomers and the mixture of 1-heptene, n-heptane, and cycloheptane on Zr-TCPE-H, Zr-TCPE-L₁-H, Zr-TCPE-L₂-H, and Zr-TCPE-DLI-H coated columns. The scale was normalized to Zr-TCPE-DLI.

L₁ indicated that the installed L₁ could block the flipping of the phenyl rings in TCPE. The above results provided a direct tool to in situ observe the flipping of the phenyl ring and its rigidification process.

## Isomer separation

It has been reported that the gas chromatographic separation performance was closely related to the size and regularity of pores in MOF-based stationary phases[39,50–53]. Thus, to investigate the influence of anisotropic rigidification of flexibility on MOF pores, high-resolution gas chromatography (GC) technique was implemented. Then, Zr-TCPE, Zr-TCPE-L₁, Zr-TCPE-L₂, Zr-TCPE-DLI, Zr-TCPE-H, Zr-TCPE-L₁-H, Zr-TCPE-L₂-H, and Zr-TCPE-DLI-H were dynamically coated on the capillary columns as stationary phases for GC separation (Supplementary Fig. 53). It can be seen from the cross-view SEM images of these columns that all the materials were uniformly coated on the inner wall and the coating process did not change the morphology of MOFs (Supplementary Fig. 54). The GC separation on the above MOF columns was feasible due to their thermal stabilities from the thermogravimetric results (Supplementary Fig. 52).

Six groups of isomers, including hexane, octane, nonane, heptene, octene, and ethyltoluene isomers, were employed to evaluate the separation ability of these MOFs (Fig. 6). The Zr-TCPE-H showed seriously low separation ability compared with the original Zr-TCPE (Supplementary Figs. 55 and 56). To further investigate the mechanism of separation differences between Zr-TCPE and Zr-TCPE-H, thermodynamic adsorption and kinetic diffusion experiments were carried out. The adsorption enthalpy (ΔH), adsorption entropy (ΔS), diffusion constant ($D_s$), and the resistance to mass transfer coefficient ($C_s$) values were calculated according to van't Hoff equation and Golay equation,

respectively (see the Supplementary Information for detail). The ΔH values for 2-methylpentane were more negative on Zr-TCPE-H coated column than that on Zr-TCPE coated column (Supplementary Fig. 57), indicating the stronger adsorption interactions between the analyte and Zr-TCPE-H[54,55]. The higher magnitude of ΔS value in the Zr-TCPE-H coated column demonstrated that the analyte had less conformational freedom and was more restricted in Zr-TCPE-H, indicating greater retention[55]. These differences can be attributed to the different pore sizes of Zr-TCPE (9.4 Å) and Zr-TCPE-H (6.8 Å). Both of the pore sizes were accessible to 2-methylpentane, while the smaller pore size of Zr-TCPE-H led to stronger van der Waals interactions between the analyte and pore walls[51]. Thus, the Zr-TCPE-H coated column presented higher thermodynamic adsorption to the analyte. The $D_s$ values for 2-methylpentane in Zr-TCPE and Zr-TCPE-H coated columns were $2.02 \times 10^{-12}$ m²/s and $6.71 \times 10^{-15}$ m²/s, respectively (Supplementary Fig. 58). The smaller diffusion constant demonstrated a lower diffusion rate for the analyte in Zr-TCPE-H. In addition, the $C_s$ value for 2-methylpentane in Zr-TCPE-H (0.25) was larger than that in Zr-TCPE ($7.10 \times 10^{-4}$). This phenomenon proved the larger mass transfer resistance in Zr-TCPE-H than that in Zr-TCPE[39,56]. All the above results manifested that the smaller pore size of Zr-TCPE-H induced strong thermodynamic interaction and slow kinetic diffusion, leading to poor separation performance with peak broadening and peak tailing. It was worth noting that the separation performance of Zr-TCPE coated column decreased under the longer heating time, proving the structure change from Zr-TCPE to Zr-TCPE-H (Fig. 6a).

Comparatively, Zr-TCPE-L₂-H retained a good separation ability to baseline separate all the isomers without peak tailing compared with Zr-TCPE-L₂ (Supplementary Figs. 59 and 60). This phenomenon

proved the enhanced stability of Zr-TCPE-$L_2$. The coordination of $L_2$ at pocket B greatly rigidified the flexibility along the *c*-axis and restricted the intramolecular motion of organic ligands. Meanwhile, the $L_2$ dangled on pocket A restricted the decrease of pores along the *a*-axis. As a result, the pore size and pore orderliness of Zr-TCPE-$L_2$ were mostly maintained after heating, leading to enhanced separation stability. However, Zr-TCPE-$L_1$-H showed fair separation ability with obvious peak tailing compared with Zr-TCPE-$L_1$ (Supplementary Figs. 61 and 62). Although the installation of $L_1$ at pocket A restricted the intermolecular motion of organic ligands on the ab plane, the intramolecular motion still existed. This intramolecular motion would result in the disordered orientation of phenyl rings in the pore channel, which led to reduced separation ability of Zr-TCPE-$L_1$-H. Note that Zr-TCPE-$L_1$ exhibited much better separation resolution towards ethyltoluene isomers than Zr-TCPE-$L_2$ and Zr-TCPE (Fig. 6b). It came from the smaller pores introduced by the pore partition on pocket A, leading to a more suitable size and shape matching between the pores of MOF and ethyltoluene isomers.

Among all eight MOF columns, Zr-TCPE-DLI demonstrated the best separation resolution for all isomers (Fig. 6, Supplementary Fig. 63, and Supplementary Table 6). Compared with Zr-TCPE, the separation resolution of nonane isomers on the Zr-TCPE-DLI coated column displayed almost no decrease with the extension of heating time, confirming the enhanced separation stability after linker installation (Supplementary Fig. 64). It was worth noting that there was almost no separation efficiency loss in Zr-TCPE-DLI-H (Supplementary Fig. 65), indicating that not only its structural flexibility was completely rigidified but also the pore partition provided suitable pores (Fig. 6). Furthermore, the Zr-TCPE-DLI-H column exhibited identical separation ability towards $C_8$ isomers with eight-times injections (Supplementary Fig. 66), suggesting good column repeatability. Besides, after storage for three months, the Zr-TCPE-DLI-H coated column still showed excellent separation resolution for alkane and alkene isomers, demonstrating improved stability of the Zr-TCPE-DLI (Supplementary Fig. 67). These different separation results further demonstrated the framework flexibility and its rigidification strongly affected and anisotropically tuned the separation properties.

In summary, we synthesized an anisotropically flexible TPE-based Zr-MOF with a non-interpenetrated scu topology. There were two types of vacant coordination pockets inside the framework. The Zr-TCPE exhibited reversible thermofluorochromism behavior due to the anisotropic flexibility. Linker installation was implemented to anisotropically rigidify the framework flexibility. By installing $L_1$ or $L_2$ at pocket A or B, respectively, the flexibility along the *b*- and *c*-axis was rigidified correspondingly. Simultaneously, the inter- and intramolecular motion of organic ligands were restricted. The complete rigidification of anisotropic flexibility and ligand motion resulted in enhanced stability and separation performance of Zr-TCPE. The rigidification of anisotropic flexibility guides the investigation of the molecular motions of TPE-based ligands and the construction of stable stationary phases and emitting materials.

## Methods

### Synthesis of Zr-TCPE
Zr-TCPE was synthesized according to the previous work[39]. The 10 mg $ZrCl_4$ and 120 μL AA were dissolved in 2 mL DMF in a 4 mL vial. The vial was heated at 100 °C for 1 h. After cooling down to room temperature, 20 μL $H_2O$, 160 μL AA, and 10 mg $H_4$TCPE ligand were added to the mixture. After sonication, the vial was heated at 120 °C for 24 h. The white product was collected by centrifugation and washed with DMF and EtOH three times, respectively, before drying at 60 °C under vacuum.

### Synthesis of Zr-TCPE-$L_1$ and Zr-TCPE-$L_2$
The Zr-TCPE-$L_1$ and Zr-TCPE-$L_2$ were synthesized according to previous report[30]. Typically, as-synthesized Zr-TCPE (about 15 mg) was

soaked in DMF (2 mL) with the addition of the secondary linkers (0.03 M, 4.6 mL DMF). The mixture was stirred at 75 °C for 24 h. The products were collected by centrifugation and soaked in fresh DMF for 3 days to remove uncoordinated linkers.

### DFT calculations
The DFT calculations were performed with the DMol3 programs of Materials Studio 19.1.0.2353. The Local Density Approximation (LDA)-PWC functional and DND 3.5 basis were employed for all calculations. During the calculation processes, the default settings embedded in Materials Studio 19.1.0.2353 were used for all calculations. The structural models of $H_4$TCPE ligands were built based on the structures of Zr-TCPE and Zr-TCPE-H, where carboxylates of linkers protonated to balance the charge. To reflect the constraints of the MOF lattice on the conformation of $H_4$TCPE ligands, the carbon atom of the carboxylate group was fixed during the geometry optimization and orbital calculation.

### $^2$H solid-state NMR experiments
The experiments were performed on a Bruker 400WB AVANCE III spectrometer at the field of 9.4 T using the solid echo pulse sequence (90°x-$\tau_1$-90°y- $\tau_2$−acquisition) with scans of 10240. $\tau_1$ was set to 150 μs, and $\tau_2$ was set to 0 to obtain the complete echo signal. $^2$H line shape simulations were carried out as a phenyl flipping model using the EXPRESS package running in MATLAB software. The quadrupolar coupling constant CQ and asymmetry parameter η of deuterium was set to 172 kHz and 0.03 according to the Pake patterns under 300 K.

## Data availability
The data that support the conclusions of this study are either presented in the paper or its Supplementary Information. The data are available from the corresponding authors upon request. Source data are provided with this paper.

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

## Acknowledgements

This work is supported by the National Natural Science Foundation of China (22174067, Z.-Y.G. and 22204078, M.X.), the Natural Science Foundation of Jiangsu Province of China (BK20220370, M.X.), Jiangsu Provincial Department of Education (22KJB150009, M.X.), the Priority Academic Program Development of Jiangsu Higher Education Institutions, State Key Laboratory of Analytical Chemistry for Life Science (SKLACLS2218, Z.-Y.G.), and the Postgraduate Research&Practice Innovation Program of Jiangsu Province (KYCX23_1684, S.-S.M.). This work is carried out with the support of Shanghai Synchrotron Radiation Facility Beamline BL17B1 (proposal 2021-NFPS-PT-006657).

## Author contributions

Z.-Y.G. conceived the idea and supervised the research. S.-S.M. and M.X. performed the synthesis, characterization, and GC experiments and discussed the results. H.-X.G. and X.-Q.K. performed the $^2$H SSNMR experiments. C.-L.C. and Y.H. performed the ultrahigh-resolution low-dose HRTEM imaging. P.-Y.C. assisted in revealing the structure of MOFs. B.D. and L.-G.X. assisted in the synthesis of deuterated $H_4$TCPE-$d_{16}$ ligands. Y.-H.G. assisted in analyzing the HAADF characterization. W.-Q.T. and W.-S.T. assisted in the GC experiments. S.Y. assisted in the DFT calculations. Z.-Y.G., S.-S.M., and M.X. discussed the experimental data and wrote the paper. All authors have approved the final version of the manuscript.

## Competing interests

The authors declare no competing interests.
