## [Peer Review File · Nature Communications]

Anisotropic Flexibility and Rigidification in a TPE-based Zr-MOFs with scu TopologyReviewers' Comments:

Reviewer #1:

Remarks to the Author:

In the manuscript, the authors synthesized an anisotropically flexible TPE-based Zr-MOFs with non-interpenetrated scu topology and utilized linker installation strategy to anisotropically rigidify the framework flexibility. The rigidified MOFs exhibited significantly enhanced stability and separation performance. This work provides an extraordinary example to precisely control the anisotropic flexibility of MOFs and offers new perspective to directionally control the inter- and intramolecular motion of organic ligands. Moreover, this work guides the design of high stable separation materials. Thus, I strongly support publication of this work after the following issues are addressed.

1. In the introduction part, several important refs about Stable Zr-MOFs are missing:
Chemical, thermal and mechanical stabilities of metal-organic frameworks. *Nat. Rev. Mater.* 2016, 1, 15018
Catalytic Zirconium/Hafnium-Based Metal-Organic Frameworks. *ACS Catal.* 2017, 7, 997-1014
2. The phenyl flipping of organic ligands exists in ftw MOFs, thus the schematic illustration in Scheme 1 and the description of ftw MOFs flexibility should be revised.
3. The authors mentioned that installing linkers into Zr-TCPE resulted in enhanced chemical stability and fluorescent stability in the Introduction Section. Please provide more data to support this statement.
4. The characterization of MOFs constructed with deuterated H4TCPE should be given to confirm the successful synthesis.
5. The authors supposed that the loss of solvents and modulators in Zr-TCPE under heating led to structural change. Please provide the data to confirm the loss of solvents.
6. To demonstrate the enhanced separation stability after linker installation, the separation performance of Zr-TCPE-DLI coated column with additional heating at 250 oC for different time should be given to compare with Zr-TCPE.
7. There are some small mistakes in the manuscript should be revised before publication. Please double check.
8. The Figures in manuscript are with low resolution. Please change.

Reviewer #2:

Remarks to the Author:

Dear Editor,

The authors report the synthesis of a Zr-based MOF using the chromophore ligand 1,1,2,2-tetra(4-carboxylphenyl)ethylene (H4TCPE) and its post-synthetic structural modification using a ligand installation strategy. This particular MOFs provided two distinct sites (termed pocket A and B) in terms of size for post synthetic ligand installation. Accordingly, the authors installed naphthalenedicarboxylate (L1) in pocket A and fumarate (L2) in pocket B. In this way they obtained material with only L1 or L2 or both auxiliary ligand in the structure.

The pristine material based on TCPE ligand (scu type structure) has been already reported (ref. #25 and 35) and studied in details in terms of its optical properties. It has been demonstrated that this materials is to an extended, flexible, as shown by powder X-ray diffraction data (PXRD) where a small shift in the observed Bragg peaks are evident. This flexibility changes that particular conformation of the chromophoric organic linker (TCPE) and in this way, interesting optical properties have been reported.

Very similar results obtained in the present work , where the as-made scu MOF when heated at 250 oC change color from white to yellow and the photoluminescent color changed from blue to light green under the 365 nm excitation. At the same time, the low angle Bragg peak is shifted at lower angle indicating a large unit cell size (Fig. 2c). Using the ligand installation procedure, the resulting

materials with L1, L2 or L1&L2 ligands show distinct behavior in terms of structural rigidification and optical properties. The results suggest an anisotropic flexible behavior upon installation of L1 and L2 ligands.

In addition to the photoluminescence study, that author used these MOFs as stationary phases in gas chromatography for the separation of different isomers from various groups including hexane, octane, nonane, heptene, octene, and ethylbenzene.

Overall, the approach to tune anisotropically the flexible behavior of this particular MOF and in turn its optical and separation properties, is an interesting work. However, the novelty is limited because significant and similar work in terms of PL properties has been already published for this particular MOF. Furthermore, the manuscript is not well written. It is very hard to read, containing speculations, and difficult to follow the discussion of the experimental results. This work is not suitable for publication in Nature Communications. I recommend a more specialized Journal after major revisions. Specific comments are provided below.

1. How do the authors know that they obtained a non-interpenetrated scu MOF instead of the known double interpenetrated version?
2. Changes in PL properties are observed after heating the material at 250 °C. It is necessary to use heat in order to observe these changes? Is it possible by post synthetic modification, for example, by removing the coordinated terminal acetate anions to trigger these changes?
3. After heating, the PL properties are restored after treatment with DMF in the presence of acetic acid at 120 °C. Is the use of temperature and acetic acid crucial to restore the properties? Additional work is needed to clarify this behavior.
4. The pXRD patterns shown in SI Fig. 23 are not discussed in detail, along with the corresponding PL properties. For example, while for individual L1 and L2 containing MOFs the pXRD between the as-made and heated material is almost unchanged, this is not the case of the solid containing both L1 and L2 where the heated solid (Zr-TCPE-DLI-H) shows a larger unit cell. This is quite interesting because the installation of both L1 and L2 is expected to rigidify the scu framework.
5. Regarding the isomer separation properties that results are confusing and not explained. For example, it is stated "The Zr-TCPE-H showed seriously low separation ability compared with the original Zr-TCPE...". This behavior is attributed to "structural deformation" but no explanation is provided, supported by experimental data. Also the statement "Zr-TCPE-L1-H showed fair separation ability with obvious peak tailing compared with Zr-TCPE-L1 (Supplementary Fig. 40 and 41), which is due to the existence of flexibility along the c-axis." is confusing.
6. The statement "It was worth noting that there was almost no separation efficiency loss in Zr-TCPE-DLI-H (Supplementary Fig. 43), indicating that not only its structural flexibility was completely rigidified" is in contrast to the pXRD data for this material (SI Fig. 23) where the Zr-TCPE-DLI-H phase shows a shift in the low angle peak towards lower 2θ values, suggesting an increase in the unit cell size and the presence of flexibility.

Reviewer #3:

Remarks to the Author:

This work describes the rigidification of an scu topology ZrMOF via linker insertion. The scu topology has two distinct pockets and here two different linkers were installed on each pocket stepwise and the resulting materials were compared. The resulting MOFs were characterized extensively and deuterated analogues of some MOFs were prepared to establish structural rigidity via solid state NMR. While the topology, the MOF itself and linker insertion for rigidification is not new, it is worth publishing the demonstration of linker rotation and also the separation performance of the MOFs. However the

following major points need to be addressed before its publication.

1- Firstly, while the authors cited the previously published TCPE MOFs with biphenyl analogs, the same ligand was also used for building scu topology Zr MOF. Authors need to cite the original work (Zirconium-based metal-organic framework gels for selective luminescence sensing RSC Adv., 2020,10, 44912-44919) and mention this in both article and SI where the synthesis of this material is mentioned.

2- While authors cited other ligand insertion studies, they missed an important one for this research which shows that insertion of the same naphlene based ligand used here into an scu Zr MOF which increased its mechanical properties, more specifically bulk modulus (Robison et al. Chem. Mater. 2020, 32, 8, 3545-3552)

3- Were the Authors able to occupy all the possible sites to install these linkers? This should be mentioned in the discussion.

4- Authors reported the BET surface area of the materials studied here with decimal points. As it was mentioned in a recent Tutorial article (DOI: <https://doi.org/10.1039/D1TA08021K>), the surface areas obtained from the BET equation is subject to error depending on the pressure region selected so it is not accurate to report surface areas with this precision as it is not that reproducible. So I recommend authors to either remove the decimal points or report the number with error bars.

5- Authors mentioned that they were not able to get the single crystal despite many trials. Their experimental section shows that they add water as well in addition to acetic acid. Water typically results in smaller particles. It could be because of the fact that they are getting other phases such as csq, in that case I recommend not using water but using a little more DMF. Not sure if this would result in diffraction quality crystals but I am nearly confident that this would give that larger crystals.

6- Authors also coated the inner surface of a column with the MOF to use it for hydrocarbon separation. However, their experimental section is not clear about exactly how the procedure was one. They mentioned it was was or x mL suspension was passed through column but they did not mention how this was done. This is an important step given that the capillary diameter is so small. I recommend authors to add more details to their experimental section.

7- Similar to above, for linker insertion, authors have cited a previous work without mentioning the amount of MOF used for each reaction. The reagents amount needs to be added to ensure others can safely reproduce your results.

Reviewers' comments:

Reviewer #1

Comments:

In the manuscript, the authors synthesized an anisotropically flexible TPE-based Zr-MOFs with non-interpenetrated scu topology and utilized linker installation strategy to anisotropically rigidify the framework flexibility. The rigidified MOFs exhibited significantly enhanced stability and separation performance. This work provides an extraordinary example to precisely control the anisotropic flexibility of MOFs and offers new perspective to directionally control the inter- and intramolecular motion of organic ligands. Moreover, this work guides the design of high stable separation materials. Thus, I strongly support publication of this work after the following issues are addressed.

Response: We highly appreciate the referee for the supportive comments and we have improved the manuscript significantly according to your following comments.

1. In the introduction part, several important refs about Stable Zr-MOFs are missing: Chemical, thermal and mechanical stabilities of meta-organic frameworks. Nat. Rev. Mater. 2016, 1, 15018. Catalytic Zirconium/Hafnium-Based Metal-Organic Frameworks. ACS Catal. 2017, 7, 997-1014

Response: Thank you for your suggestion. We have added the references in the manuscript. Please see **Page 9, right column, Reference 22 and 23** in the manuscript.

2. The phenyl flipping of organic ligands exists in ftw MOFs, thus the schematic illustration in Scheme 1 and the description of ftw MOFs flexibility should be revised.

Response: Thank you for your suggestion. We have revised the incorrect description. We have changed “the structure is rigid, in which the organic ligands are firmly and identically immobilized into the framework along a, b, and c directions.” to “although the phenyl

flipping of organic ligands exists, the organic ligands are firmly and identically immobilized into the framework along a, b, and c directions, and the framework is rigid.” Please see Page 1, left column, second paragraph in the manuscript.

We have revised the incorrect illustration in Scheme. Please see Page 2, Scheme 1 in the manuscript.

3. The authors mentioned that installing linkers into Zr-TCPE resulted in enhanced chemical stability and fluorescent stability in the Introduction Section. Please provide more data to support this statement.

Response: Thank you for pointing this out. We have added the PXRD patterns of Zr-TCPE-DLI immersing in aqueous solutions with different pH values to support enhanced chemical stability. Please see Page S39, Supplementary Figure 32 in Supplementary Information. We have added “The PXRD patterns of Zr-TCPE-DLI immersing in aqueous solutions with different pH values exhibited no obvious difference. Besides, the crystallinity of Zr-TCPE-DLI was well maintained after 15 months of storage, which further indicated the enhanced chemical stability.” Please see Page 5, right column, second paragraph in the manuscript.

Supplementary Figure 32. The PXRD patterns of (a) Zr-TCPE-DLI immersing in aqueous solutions with different pH values for 24 h and (b) Zr-TCPE-DLI after 15 months of storage. The crystallinity of Zr-TCPE-DLI was well maintained, indicating enhanced stability.

4. *The characterization of MOFs constructed with deuterated H₄TCPE should be given to confirm the successful synthesis.*

Response: Thank you for pointing this out. We have added the PXRD patterns, TEM images, and ¹H NMR spectroscopy of MOFs constructed with deuterated H₄TCPE to confirm the successful synthesis. Please see Page S46-48, Supplementary Figure 42-45 in

Supplementary Information. We have added “The PXRD patterns, TEM images, and ^1H NMR spectra confirmed the successful synthesis of Zr-TCPE, Zr-TCPE-L₁, and Zr-TCPE-L₂ constructed with H₄TCPE-d₁₆” in the manuscript. Please see Page 7, left column, the first paragraph in the manuscript.

Supplementary Figure 42. The PXRD patterns of Zr-TCPE, Zr-TCPE -L₁, and Zr-TCPE-L₂ constructed with deuterated H₄TCPE.

Supplementary Figure 43. The TEM images of (a,b) Zr-TCPE, (c,d) Zr-TCPE-L₁, and (e,f) Zr-TCPE-L₂ constructed with deuterated H₄TCPE.

Supplementary Figure 44. The ^1H NMR spectroscopy of digested Zr-TCPE-L₁ constructed with deuterated H₄TCPE.

Supplementary Figure 45. The ^1H NMR spectroscopy of digested Zr-TCPE-L₂ constructed with deuterated H₄TCPE.

5. The authors supposed that the loss of solvents and modulators in Zr-TCPE under heating led to structural change. Please provide the data to confirm the loss of solvents.

Response: Thank you for pointing this out. We have revised the ^1H NMR spectra of Zr-TCPE and Zr-TCPE-*H* to confirm the loss of solvents. We have revised “The ^1H NMR result of the digested Zr-TCPE confirmed the existence of TCPE ligands and acetic acid in the framework” to “The ^1H NMR result of the digested Zr-TCPE confirmed the existence of TCPE ligands, DMF solvents, and acetic acid in the framework.” Please see Page 3, right column, second paragraph in the manuscript. The ratio of TCPE: AA: DMF was 1: 0.5: 0.4 in Zr-TCPE, while the ratio was 1: 0.3: 0.05 in Zr-TCPE-*H*. These results proved the loss of solvents in Zr-TCPE under heating. Please see Page S13, Supplementary Figure 7, and Page S17, Supplementary Figure S11 in Supplementary Information.

Supplementary Figure 7. The ^1H NMR spectroscopy of digested Zr-TCPE (TCPE: AA: DMF = 1: 0.5: 0.4).

Supplementary Figure 11. The ^1H NMR spectroscopy of digested Zr-TCPE-*H* (TCPE: AA: DMF = 1: 0.3: 0.05). The amount of DMF in Zr-TCPE decreased apparently after heating treatment.

6. To demonstrate the enhanced separation stability after linker installation, the separation performance of Zr-TCPE-DLI coated column with additional heating at 250 °C for different time should be given to compare with Zr-TCPE.

Response: Thank you for your suggestion. We have added the separation resolution of nonane isomers on Zr-TCPE-DLI coated column with additional heating at 250 °C for 4 h, 6h, and 12 h. Please see Page S63, Supplementary Figure 58 in Supplementary Information. We have added “Compared with Zr-TCPE, the separation resolution of nonane isomers on the Zr-TCPE-DLI coated column displayed almost no difference with the extension of heating time, confirming the enhanced separation stability after linker installation” in the manuscript. Please see Page 8, left column, the first paragraph in the manuscript.

Supplementary Figure 58. (a) The gas chromatograms on Zr-TCPE-DLI coated column for the separation of nonane isomers with additional heating at 250 °C for 4 h, 6 h, and 12 h. (b) The separation resolution of nonane isomers on Zr-TCPE-DLI coated column with additional heating at 250 °C for 4 h, 6 h, and 12 h.

7. *There are some small mistakes in the manuscript should be revised before publication. Please double check.*

Response: Thank you for pointing this out. We have carefully checked the manuscript. We have revised some confusing expressions and revised mistakes. For example, we have revised “Meanwhile, with the dual linker installation (DLI), the Zr-TCPE-DLI was also obtained through the sequential synthetic method first with L₂ then L₁” to “Meanwhile,

with the dual linker installation (DLI), the Zr-TCPE-DLI was also obtained, in which the installation order is installing L_2 first and then L_1 ” in the manuscript. Please see Page 5, right column, the first paragraph in the manuscript.

8. The Figures in manuscript are with low resolution. Please change.

Response: Thank you for pointing this out. We have replaced all the Figures in the manuscript with high resolution.

Reviewer #2

Comments:

The authors report the synthesis of a Zr-based MOF using the chromophore ligand 1,1,2,2-tetra(4-carboxylphenyl)ethylene (H₄TCPE) and its post-synthetic structural modification using a ligand installation strategy. This particular MOFs provided two distinct sites (termed pocket A and B) in terms of size for post synthetic ligand installation. Accordingly, the authors installed naphthalenedicarboxylate (L₁) in pocket A and fumarate (L₂) in pocket B. In this way they obtained material with only L₁ or L₂ or both auxiliary ligand in the structure.

The pristine material based on TCPE ligand (scu type structure) has been already reported (ref. #25 and 35) and studied in details in terms of its optical properties. It has been demonstrated that this material is to an extended, flexible, as shown by powder X-ray diffraction data (PXRD) where a small shift in the observed Bragg peaks are evident. This flexibility changes that particular conformation of the chromophoric organic linker (TCPE) and in this way, interesting optical properties have been reported.

Very similar results obtained in the present work, where the as-made scu MOF when heated at 250 °C change color from white to yellow and the photoluminescent color changed from blue to light green under the 365 nm excitation. At the same time, the low angle Bragg peak is shifted at lower angle indicating a large unit cell size (Fig. 2c). Using the ligand installation procedure, the resulting materials with L₁, L₂ or L₁&L₂ ligands show distinct behavior in terms of structural rigidification and optical properties. The results suggest an anisotropic flexible behavior upon installation of L₁ and L₂ ligands. In addition to the photoluminescence study, that author used these MOFs as stationary phases in gas chromatography for the separation of different isomers from various groups including hexane, octane, nonane, heptene, octene, and ethylbenzene.

Overall, the approach to tune anisotropically the flexible behavior of this particular MOF and in turn its optical and separation properties, is an interesting work. However,

the novelty is limited because significant and similar work in terms of PL properties has been already published for this particular MOF.

Response: Thank you for your comments. We have carefully read these comments and understand that the main consideration is the novelty. We apologize for not fully emphasizing our novelty which is different from theirs', leading to the misunderstanding. Indeed, we have carefully read these two papers and fully evaluated the difference between these materials and our Zr-TCPE before the submission. We still believe our work investigates the anisotropic flexible TPE-based *scu* MOF from a special perspective with quite different phenomena. The differences between our work and others are listed below.

First, in your opinion, this Zr-TCPE (*scu* type) synthesized in our manuscript has been already reported by ref. #25 (*Chem. Mater.* 2019, 31, 5550-5557). However, in fact, for Zr-ETTC in ref. #25, the organic building block was different from the Zr-TCPE in our work. ETTC was larger than TCPE. Thus, Zr-ETTC showed a 2-fold interpenetrated structure, while our Zr-TCPE was non-interpenetrated, which was proved by the HAADF images. In ref. #25, it was unable to separately control the framework flexibility of Zr-ETTC and the framework interpenetration, thus, the PL changes could not directly reflect the conformational changes of organic ligands. While our work clearly controlled the anisotropic framework flexibility and investigated intermolecular and intramolecular motions of TCPE ligands. **Second**, ref. #35 (*J. Am. Chem. Soc.* 2023, 145, 1072-1082) was published on January 2023 during the preparation of our manuscript. We read this paper just two weeks before the submission to Nature Communications. Although the structure of Zr-TCPE in our work is indeed coincidentally similar to Form 2 in ref. #35, our synthesis method and especially the major scientific findings are quite different from the reference. Thus, we have confidently and objectively cited this paper in our previous submission. We understand your consideration, but we still would like to show the difference in major discoveries between our work and ref. #35.

The mechanism and modulation methods for PL changes are different. We agree with

the comment that PL changes in TPE-based MOFs are common phenomena. Thus, the special chemical modulation method and unique mechanism were crucial to the novelty. It is still a hot research topic. At least, dozens of papers published in top journals to develop new modulation strategies or reveal new mechanisms in this specific area. In ref. #35, the different PL spectra were observed for a series of Zr-TCPE samples with different amounts of modulators in the synthesis procedures. **The blue-shift PL changes were due to the phase transition between two different forms of Zr-TCPE, not the flexibility of pure scu Zr-TCPE.** The linker substitution defects during synthesis are the key to the generation of this PL change. In our work, the anisotropic stimuli-responsive emission property of Zr-TCPE was discovered for the individual sample under different temperatures. The maximum emission wavelength exhibited apparent red-shift PL properties. We contributed this PL change to the inter- and intra-molecular motion of organic ligands and performed a newly developed modulation strategy to anisotropically control each motion.

The anisotropic flexibility and rigidification, not the PL changes, are the notable merits of our work. PL change of Zr-TCPE is a basic characterization method for us to illustrate the status of the ligand motions, but not the whole part of it. Both the anisotropic flexibility of Zr-TCPE and its rigidification are first reported in our work. We have employed many characterization methods to support this observation, such as HAADF images, XRD patterns, Raman spectra, DFT calculations, rational design of installed linkers, fluorescence emission spectra, stability test, *in-situ* ^2H solid-state NMR, and improved GC separation performance. Thus, the re-evaluation of the novelty from the view of the anisotropic flexibility and its rigidification, but not solely from the PL characterization data, is suggested.

Our work and ref. #35 are independent. This statement is supported by many experimental details. For example, the synthesis methods and structural characterization methods are different. We obtained the Zr-TCPE through a two-step synthesis method based on our previous work (*ACS Cent. Sci.* 2022, 8, 184-191) with only AA modulator,

while, for ref. #35, it was synthesized in a conventional one-pot solvothermal reaction with different modulators (FA, AA, or BA). The Zr-TCPE structure was elucidated from the combination of HAADF images, XRD measurement, and Pawley refinement in our manuscript. In contrast, ref. #35 employed only XRD and Rietveld refinement to solve the structure. Besides, the obtained crystal parameters in our work are slightly different from the reported ones.

We have added the following discussion in the manuscript to illustrate the differences and emphasize our novelty:

We have revised “Recently, the interpenetrated **scu** TPE-based Zr-MOF have been synthesized as stimuli-responsive emitters with solvato-, thermo-, and piezofluorochromism” to “The **scu** Zr-MOF (LIFM-114) constructed with ETTC ligand has been synthesized with a 2-fold interpenetrated structure, which made it complex to analyze the different motions of organic ligands” in the manuscript. Please see Page 1, right column, last paragraph in the manuscript.

We have added “Although, very recently, a non-interpenetrated **scu** Zr-MOF has been reported, the inter- and intramolecular motions of organic ligands have yet not been investigated” in the manuscript. Please see Page 2, left column, the first paragraph in the manuscript.

We have added “Thus, it is possible to control the anisotropic flexibility of **scu** MOFs by installing different second linkers at different positions” in the manuscript. Please see Page 2, left column, second paragraph in the manuscript.

We have added “The crystallographic parameters were slightly different from the reported **scu** TPE-based MOFs” in the manuscript. Please see Page 2, right column, last paragraph in the manuscript.

We have provided more experiments to strengthen our conclusion. We have added the Raman spectra of Zr-TCPE and Zr-TCPE-*H*. Please see Page S21, Supplementary Figure 16 in Supplementary Information. We have added “The Raman peak at 412 cm⁻¹ was

assigned to the stretching modes of Zr-O (carboxylate), and the peak at 1125 cm^{-1} was assigned to the stretching vibration of the C-C bond formed between ethylene and phenyl ring. Another obvious Raman peak at 1607 cm^{-1} was assigned to the stretching vibration of the C=C band from phenyl rings and ethylene core. Compared with Zr-TCPE, neither Raman peaks disappeared nor new Raman peaks appeared in Zr-TCPE-*H*, indicating unchanged chemical composition in Zr-TCPE after heating” in the manuscript. Please see Page 4, right column, the first paragraph in the manuscript.

Supplementary Figure 16. The Raman spectra of Zr-TCPE and Zr-TCPE-*H*.

We have added the following references:

1. Guo, X. et al. Stimuli-Responsive Luminescent Properties of Tetraphenylethene-Based Strontium and Cobalt Metal-Organic Frameworks. *Angew. Chem. Int. Ed.* **59**, 19716-19721 (2020).
2. Liu, X., Li, A., Xu, W., Ma, Z. & Jia, X. Pressure-induced emission band separation of the hybridized local and charge transfer excited state in a TPE-based crystal. *Phys. Chem. Chem. Phys.* **20**, 13249-13254 (2018).
3. Gao, Z. et al. Enhanced Sensitivity and Piezochromic Contrast through Single-Direction Extension of Molecular Structure. *Chem. Eur. J.* **23**, 773-777 (2017).

Furthermore, the manuscript is not well written. It is very hard to read, containing

speculations, and difficult to follow the discussion of the experimental results. This work is not suitable for publication in Nature Communications. I recommend a more specialized Journal after major revisions. Specific comments are provided below.

Response: Thank you for your comments. We have deleted some confusing expressions and speculations and added a more precise discussion of experimental results.

We have added “The intramolecular motion of TCPE ligands in Zr-TCPE was associated with the framework change along the c-axis, indicating the framework flexibility along the c-axis. The close stacking of TCPE ligands in Zr-TCPE corresponded to the framework change on the ab plane” in the manuscript. Please see **Page 3, right column, the last paragraph** in the manuscript.

We have added “All the results demonstrated the framework change along the b-axis after heating treatment, indicating the framework flexibility along the b-axis” in the manuscript. Please see **Page 4, left column, the first paragraph** in the manuscript.

We have added “In addition, the main pore size of Zr-TCPE-H was 6.8 Å, smaller than that of Zr-TCPE (Supplementary Fig. 14). The reduced pore size proved the shrinkage along the a-axis, indicating the framework flexibility along the a-axis. The above experiments and characterizations demonstrated the anisotropic flexibility of Zr-TCPE (Fig. 2e)” in the manuscript. Please see **Page 4, left column, the first paragraph** in the manuscript. Please see **Page S20, Supplementary Figure 14** in Supplementary Information.

Supplementary Figure 14. (a) The N₂ adsorption-desorption isotherms and (b) pore size distribution of Zr-TCPE-*H* measured at 77K.

We have revised “Meanwhile, with the dual linker installation (DLI), the Zr-TCPE-DLI was also obtained through the sequential synthetic method first with L₂ then L₁” to “Meanwhile, with the dual linker installation (DLI), the Zr-TCPE-DLI was also obtained, in which the installation order is installing L₂ first and then L₁” in the manuscript. Please see Page 5, right column, the first paragraph in the manuscript.

We have revised “Thus, we supposed that pocket A was fully occupied by L₁ in Zr-TCPE-L₁ as the possibility of L₁ at pocket B was ruled out due to the size mismatching” to “Because of the size mismatching between the space of pocket B and the length of L₁, the possibility of L₁ coordinated on pocket B was ruled out. Thus, pocket A was fully occupied by L₁ in Zr-TCPE-L₁” in the manuscript. Please see Page 5, left column, second paragraph

in the manuscript.

We have revised “We supposed that the steric hindrance of pocket B in the original MOF prevented full coordination at pocket B” to “It resulted from the steric hindrance of pocket B in the original MOF that prevented full coordination at pocket B” in the manuscript. Please see Page 5, left column, second paragraph in the manuscript.

We have revised “We supposed that because L₂ also possessed the ability to coordinate on the metal sites at pocket A” to “It was because L₂ possessed the ability to coordinate on the metal sites at pocket A” in the manuscript. Please see Page 6, left column, the first paragraph in the manuscript.

We have revised “We supposed that the installation of L₁ at pocket A rigidified the flexibility and restricted the intermolecular stacking between adjacent organic ligands along the b-axis” to “The installation of L₁ at pocket A rigidified the flexibility and restricted the intermolecular stacking between adjacent organic ligands on the ab plane” in the manuscript. Please see Page 6, left column, second paragraph in the manuscript.

We have re-witted the discussion of GC separation experiments and added the discussion of the mechanism of separation difference. Please see Page 7, Isomer Separation, in the manuscript.

1. How do the authors know that they obtained a non-interpenetrated scu MOF instead of the known double interpenetrated version?

Response: Thank you for pointing this out. We have added the following discussion in Supplementary Information to support our statement. Please see Page S8, Supplementary Figure 3 in Supplementary Information.

We have added “On the one hand, due to the failure of single crystal synthesis, the structure of Zr-TCPE was analyzed by Pawley refinement. The low R-value indicated the refined profile matched the experimental PXRD pattern very well. The refinement results revealed that each TCPE ligand was connected to four Zr₆ clusters and each Zr₆ cluster was

coordinated with eight TCPE ligands. Then, the non-interpenetrated **scu** coordination structure was generated in Zr-TCPE. On the other hand, the distance of adjacent Zr_6 clusters was 1.77 nm along the a-axis in the simulated non-interpenetrated structure, which was consistent with the HAADF image measurements ($d=1.79$ nm). However, the distance of adjacent Zr_6 clusters in a double-interpenetrated structure would be 0.89 nm, which was not found in any of the HAADF image measurements. Meanwhile, the distance of adjacent Zr_6 clusters was 3.02 nm along the b-axis in the simulated non-interpenetrated structure, which was also consistent with the HAADF image measurements ($d=3.04$ nm). Therefore, we assigned a non-interpenetrated **scu** structure to Zr-TCPE” Please see Page S11 in Supplementary Information.

We have revised “The measured distance of adjacent Zr_6 clusters was around 1.77 nm, which was consistent with the simulated value along the a-axis. Along its perpendicular direction, the b axis, the distance between two adjacent Zr_6 clusters was hard to observe directly” to “Due to the limitation of the resolution, the distance between two adjacent Zr_6 clusters along the b-axis was hard to observe directly because of the interference from Zr_6 clusters along the a-axis. Thus, the denoised images were acquired by an inverse-FFT process after applying a periodic mask to the FFT pattern for measuring the distance of Zr_6 clusters. The measured distance of adjacent Zr_6 clusters was around 1.79 nm and 3.04 nm (Supplementary Fig. 4). These values were consistent with the simulated value along the a-axis and b-axis, respectively, in a non-interpenetrated structure, but differed from the distance of adjacent Zr_6 clusters in a double-interpenetrated structure (0.89 nm and 1.51 nm, see Supplementary Fig. 3). The results further proved that the Zr-TCPE possessed a non-interpenetrated structure.” in the manuscript. Please see Page 3, left column, the last paragraph in the manuscript.

We have added “Besides, the measured distance of adjacent Zr_6 clusters along the b-axis had also increased from 3.04 nm to 3.25 nm” in the manuscript. Please see Page 4, right column, the first paragraph in the manuscript.

Supplementary Figure 3. The distance of adjacent Zr_6 clusters along the a-axis and b-axis in (a) non-interpenetrated scu topology and (b) double-interpenetrated scu topology.

Supplementary Figure 5. (a,d,g) The HAADF images of Zr-TCPE. (b,e,h) The denoised HAADF images of the red square region. The denoised image was acquired by an inverse-FFT process after applying a periodic mask to the FFT pattern. (c,f, i) The measured distance of adjacent Zr_6 clusters was along the a-axis (labeled in yellow) and the b-axis (labeled in blue). The distance of adjacent Zr_6 clusters along the a-axis was around 1.79 nm, which was similar to the simulated value ($d=1.77$ nm). The distance of adjacent Zr_6 clusters along the b-axis was around 3.04 nm, which was similar to the simulated value ($d=3.02$ nm).

Supplementary Figure 13. (a,d,g) The HAADF images of Zr-TCPE-H. (b,e,h) The denoised HAADF images of the red square region. The denoised image was acquired by an inverse-FFT process after applying a periodic mask to the FFT pattern. (c,f, i) The measured distance of adjacent Zr_6 clusters was along the a-axis (labeled in yellow) and the b-axis (labeled in blue). The distance of adjacent Zr_6 clusters along the a-axis and b-axis was around 1.79 nm and 3.25 nm, respectively.

2. Changes in PL properties are observed after heating the material at 250 °C. It is necessary to use heat in order to observe these changes? Is it possible by post synthetic

modification, for example, by removing the coordinated terminal acetate anions to trigger these changes?

Response: Thank you for your suggestion. We have added the experiments to remove the coordinated terminal acetate anions by treating the as-synthesized Zr-TCPE with 8 M HCl in DMF at 100 °C for 24 h according to the previous works (*J. Am. Chem. Soc.* **2013**, 135, 10294-10297; *J. Am. Chem. Soc.* **2020**, 142, 21110-21121). However, the removal of acetate acid did not trigger PL changes. In addition, the removal of acetic acid and subsequent heating at different temperatures, such as 120 °C and 150 °C, resulted in slight fluorescence changes. The above results prove that the structural change caused by heating at 250 °C plays a significant role in PL changes.

The material (Zr-TCPE-HCl) was activated at 120 °C for 6 h. The ¹H NMR spectroscopy demonstrates that the ratio of TCPE: AA: DMF is 1: 0.07: 0.4. This result indicates the successful removal of coordinated terminal acetate. The maximum emission wavelength of this activated material was 466 nm. Increasing the activation time to 18 h, the maximum emission wavelength did not change (466 nm). Further increasing the activation temperature to 150 °C, the color of the activated material changed from white to yellow. The maximum emission wavelength of this activated material is 470 nm and the ratio of TCPE: AA: DMF is 1: 0.06: 0.1. These results indicated that removing the acetic acid could not trigger the PL changes, further proving the structural change induced by heating at 250 °C is important to PL changes.

Figure R1. The ^1H NMR spectroscopy of Zr-TCPE-HCl after activation at 120 °C for 6 h (TCPE: AA: DMF = 1: 0.07: 0.4).

Figure R2. The ^1H NMR spectroscopy of Zr-TCPE-HCl after activation at 120 °C for 18 h (TCPE: AA: DMF = 1: 0.07: 0.3).

Figure R3. The ^1H NMR spectroscopy of Zr-TCPE-HCl after activation at 150 °C for 6 h (TCPE: AA: DMF = 1: 0.06: 0.1).

Figure R4. The fluorescence spectra of Zr-TCPE-HCl after activation at different temperatures. These MOFs were dispersed in methanol at a concentration of 40 $\mu\text{g/mL}$ ($\lambda_{\text{ex}}=340$ nm).

3. *After heating, the PL properties are restored after treatment with DMF in the presence of acetic acid at 120 °C. Is the use of temperature and acetic acid crucial to restore the properties? Additional work is needed to clarify this behavior.*

Response: Thank you for your suggestion. We have added the experiments to restore the PL properties of Zr-TCPE-*H* with different amounts of acetic acid at different temperatures. We have added “When treated at room temperature, the PXRD patterns and PL properties of Zr-TCPE-*H* did not restore, despite the addition of DMF and acetic acid. When the temperature was raised to 120 °C, the PXRD pattern and fluorescence emission were restored even without the addition of acetic acid. These experiments verified the importance of temperature in restoring PL properties. Besides, at 120 °C, DMF could hydrolyze into formic acid. Thus, acid coordination is inevitable during structural restoration.” Please see Page S23-26, Section S6 in Supplementary Information.

We have revised “Then, these heated materials were treated in DMF solution with acetic acid at 120 °C for 12 h. After this treatment, the PXRD patterns and fluorescence emission recovered” to “Then, these heated materials were treated in DMF solution with acetic acid under different temperatures for 12 h (see Supplementary Information for detail). The use of temperature is important to restore the structure. At 100 °C or 120 °C, the PXRD patterns and fluorescence emission were completely restored with the addition of acetic acid (Fig. 2d and Supplementary Fig. 17-19)” in the manuscript. Please see Page 4, right column, second paragraph in the manuscript.

We have added “The 10 mg Zr-TCPE-*H* was dispersed in 2 mL DMF in a glass vial with the addition of 0/200/400 μL AA. Then the vial was heated at 120 °C for 12 h in an oven. After cooling to room temperature, the product was collected by configuration and washed with DMF and EtOH three times, respectively, before drying under vacuum.” The PXRD patterns and fluorescence emission spectra of these materials were recorded. Please see Page S24, Supplementary Figure 17 in Supplementary Information.

Supplementary Figure 17. (a) The PXRD patterns and (b) the fluorescence spectra of Zr-TCPE-*H* after treatment in DMF solution with different amounts of acetic acid at 120 $^{\circ}\text{C}$ for 12 h. These MOFs were dispersed in methanol at a concentration of 40 $\mu\text{g}/\text{mL}$ ($\lambda_{\text{ex}}=340$ nm). After treatment at 120 $^{\circ}\text{C}$ without acetic acid, the maximum emission wavelength was 472 nm.

We have added “The 10 mg Zr-TCPE-*H* was dispersed in 2 mL DMF in a glass vial with the addition of 200/400 μL AA. Then the vial was heated at 100 $^{\circ}\text{C}$ for 12 h in an oven. After cooling to room temperature, the product was collected by configuration and washed with DMF and EtOH three times, respectively, before drying under vacuum.” The PXRD patterns and fluorescence emission spectra of these materials were recorded. Please see Page S25, Supplementary Figure 18 in Supplementary Information.

Supplementary Figure 18. (a) The PXRD patterns and (b) the fluorescence spectra of Zr-TCPE-*H* after treatment in DMF solution with different amounts of acetic acid at 100 $^{\circ}\text{C}$ for 12 h. These MOFs were dispersed in methanol at a concentration of 40 $\mu\text{g}/\text{mL}$ ($\lambda_{\text{ex}}=340$ nm). The maximum emission wavelengths of Zr-TCPE-*H* after treatment with 200 μL AA

and 400 μL AA were 466 nm and 472 nm, respectively.

We have added “The 10 mg Zr-TCPE-*H* was dispersed in 2 mL DMF in a glass vial with the addition of 200/400 μL AA. Then the vial was kept at room temperature for 12 h. The product was collected by configuration and washed with DMF and EtOH three times, respectively, before drying under vacuum.” Please see Page S26, Supplementary Figure 19 in Supplementary Information.

Supplementary Figure 19. (a) The PXRD patterns and (b) the fluorescence spectra of Zr-TCPE-*H* after treatment in DMF solution with different amounts of acetic acid at room temperature for 12 h. These MOFs were dispersed in methanol at a concentration of 40 $\mu\text{g}/\text{mL}$ ($\lambda_{\text{ex}}=340$ nm). Neither PXRD patterns nor fluorescence spectra of Zr-TCPE-*H* were restored.

4. The pxd patterns shown in SI Fig. 23 are not discussed in detail, along with the corresponding PL properties. For example, while for individual L1 and L2 containing MOFs the pxd between the as-made and heated material is almost unchanged, this is not the case of the solid containing both L1 and L2 where the heated solid (Zr-TCPE-DLI-H) shows a larger unit cell. This is quite interesting because the installation of both L1 and L2 is expected to rigidify the scu framework.

Response: Thank you for pointing this out. We apologize for the mistake in the data processing. We have carefully rechecked the patterns and found that the PXRD pattern of Zr-TCPE-DLI-H was incorrectly assigned. We have revised the PXRD patterns in Supplementary Information. Please see Page S38, Supplementary Figure 30 in Supplementary Information.

Supplementary Figure 30. The PXRD patterns of Zr-TCPE-L₁, Zr-TCPE-L₁-H, Zr-TCPE-L₂, Zr-TCPE-L₂-H, Zr-TCPE-DLI, and Zr-TCPE-DLI-H.

We have repeatedly tested the PXRD pattern of Zr-TCPE-DLI-H with different substrates. The result indicated that there was no shift in PXRD patterns. In addition, we have resynthesized Zr-TCPE-DLI and Zr-TCPE-DLI-H and then repeated the test of their PXRD patterns to confirm again. It can be confirmed that the PXRD pattern of Zr-TCPE-DLI-H has no shift.

Figure R5. (a) The PXRD patterns of Zr-TCPE-DLI and Zr-TCPE-DLI-*H* retested with different substrates. (b) The PXRD patterns of re-synthesized Zr-TCPE-DLI and re-synthesized Zr-TCPE-DLI-*H* retested with different substrates.

5. Regarding, the isomer separation properties that results are confusing and not explained. For example, it is state “The Zr-TCPE-*H* showed seriously low separation ability compared with the original Zr-TCPE...”. This behavior is attributed to “structural deformation” but no explanation is provided, supported by experimental data. Also the statement “Zr-TCPE-*L1-H* showed fair separation ability with obvious peak tailing compared with Zr-TCPE-*L1* (Supplementary Fig. 40 and 41), which is due to the existence of flexibility along the *c*-axis.” is confusing.

Response: Thank you for your suggestion. We have added experiments and explanations in the discussion of the isomer separation properties.

We have revised “To investigate the influence of anisotropic rigidification of flexibility on MOF separation properties, high-resolution gas chromatography (GC) technique was implemented” to “It has been reported that the gas chromatographic separation performance was closely related to the size and regularity of pores in MOF-based stationary phases.^{39, 45-48} Thus, to investigate the influence of anisotropic rigidification of flexibility on MOF pores, high-resolution gas chromatography (GC)

technique was implemented” in the manuscript. Please see Page 7, left column, second paragraph in the manuscript.

We have added “To further investigate the mechanism of separation differences between Zr-TCPE and Zr-TCPE-*H*, thermodynamic adsorption and kinetic diffusion experiments were carried out. The adsorption enthalpy (ΔH), adsorption entropy (ΔS), diffusion constant (D_s), and the resistance to mass transfer coefficient (C_m) values were calculated according to van’t Hoff equation and Golay equation, respectively (see the Supplementary Information for detail). The ΔH values for 2-methylpentane were more negative on Zr-TCPE-*H* coated column than that on Zr-TCPE coated column (Supplementary Fig. 51), indicating the stronger adsorption interactions between the analyte and Zr-TCPE-*H*. The higher magnitude of ΔS value in the Zr-TCPE-*H* coated column demonstrated that the analyte had less conformational freedom and was more restricted in Zr-TCPE-*H*, indicating greater retention. These differences can be attributed to the different pore sizes of Zr-TCPE (9.4 Å) and Zr-TCPE-*H* (6.8 Å). Both of the pore sizes were accessible to 2-methylpentane, while the smaller pore size of Zr-TCPE-*H* led to stronger van der Waals interactions between the analyte and pore walls. Thus, the Zr-TCPE-*H* coated column presented higher thermodynamic adsorption to the analyte. The D_s values for 2-methylpentane in Zr-TCPE and Zr-TCPE-*H* coated columns were 2.02×10^{-12} m²/s and 6.71×10^{-15} m²/s, respectively (Supplementary Fig. 52). The smaller diffusion constant demonstrated a lower diffusion rate for the analyte in Zr-TCPE-*H*. In addition, the C_m values for 2-methylpentane in Zr-TCPE-*H* (0.25) was larger than that in Zr-TCPE (7.10×10^{-4}). This phenomenon proved the larger mass transfer resistance in Zr-TCPE-*H* than that in Zr-TCPE. All the above results manifested that the smaller pore size of Zr-TCPE-*H* induced strong thermodynamic interaction and slow kinetic diffusion, leading to the poor separation performance with peak broadening and peak tailing” in the manuscript. Please see Page 7, right column, the first paragraph in the manuscript. Please see Page S55-

58, Supplementary Figure 51-52 in Supplementary Information.

We have added the following references:

4. Chang, N., Gu, Z.-Y. & Yan, X.-P. Zeolitic imidazolate framework-8 nanocrystal coated capillary for molecular sieving of branched alkanes from linear alkanes along with high-resolution chromatographic separation of linear alkanes. *J. Am. Chem. Soc.* **132**, 13645-13647 (2010).
5. Chang, N. & Yan, X.-P. Exploring reverse shape selectivity and molecular sieving effect of metal-organic framework UIO-66 coated capillary column for gas chromatographic separation. *J. Chromatogr. A*, **1257**, 116-124 (2012).
6. Tao, Z.-R. et al. Untwisted restacking of two-dimensional metal-organic framework nanosheets for highly selective isomer separations. *Nat. Commun.* **10**, 2911 (2019).
7. Tang, W.-Q. et al. Controlling the stacking modes of metal-organic framework nanosheets through host-guest noncovalent interactions. *Angew. Chem. Int. Ed.* **60**, 6920-6925 (2021).
8. Xu, M. et al. Homogeneously mixing different metal-organic framework structures in single nanocrystals through forming solid solutions. *ACS Cent. Sci.* **8**, 184-191 (2022).
9. Natraj, A. et al. Single-crystalline imine-linked two-dimensional covalent organic frameworks separate benzene and cyclohexane efficiently. *J. Am. Chem. Soc.* **144**, 19813-19824 (2022).
10. Yusuf, K. et al. Inverse gas chromatography demonstrates the crystallinity-dependent physicochemical properties of two-dimensional covalent organic framework stationary phases. *Chem. Mater.* **35**, 1691-1701 (2023).
11. Meng, S.-S. et al. Enhancing separation abilities of “low-performance” metal-organic framework stationary phases through size control. *Anal. Chem.* **94**, 14251-14256 (2022).

We have added “The thermodynamic adsorption enthalpy (ΔH) was calculated from the van't Hoff equation (A):

$$\ln k' = -\frac{\Delta H}{RT} + \frac{\Delta S}{R} + \ln \emptyset \quad (\text{A})$$

Here, k' is the retention factor, R is the gas constant ($R= 8.314 \text{ J}\cdot\text{mol}^{-1}\cdot\text{K}^{-1}$), T is the absolute temperature, and \emptyset is the phase ratio (the ratio of the volume of the stationary phase (V_s) to that of the mobile phase (V_m)). To obtain \emptyset , V_s was calculated from the thickness of MOFs coated on the capillary column, while V_m was calculated from the column internal volume subtracting V_s .

k' is calculated from equation (B):

$$k' = \frac{t-t_0}{t_0} \quad (\text{B})$$

t is the retention time of the analyte, and t_0 is the retention time of an unretained compound on the column.” Please see Page S55, Section S14 in Supplementary Information.

Supplementary Figure 51. The van't Hoff plots of 2-methylpentane on (a) Zr-TCPE and (b) Zr-TCPE-*H* coated columns. The thermodynamic adsorption enthalpy (ΔH) of 2-methylpentane on Zr-TCPE and Zr-TCPE-*H* coated column was -51.89 kJ/mol and -58.81 kJ/mol, respectively.

We have added “The kinetic diffusion constant (D_s) was calculated from the Golay equation:

$$H = \frac{2D_g}{u} + \frac{1+6k+11k^2}{24(1+k)^2} \times \frac{r^2}{D_g} \times u + \frac{2}{3} \times \frac{k}{(1+k)^2} \times \frac{d_f^2}{D_s} \times u \quad (C)$$

Here, D_g is the diffusion constant of the analyte in the gas phase, D_s is the diffusion constant of the analyte in the stationary phase, r is the radius of the capillary column ($r=125$ mm), d is the thickness of the stationary phase ($d=100$ nm), u is the linear velocity of the carrier gas, H is the height equivalent of the theoretical plate.

The resistance to mass transfer coefficient (C_m) is calculated based on equation (D):

$$C_m = \frac{2}{3} \times \frac{k}{(1+k)^2} \times \frac{d_f^2}{D_s} \quad (D)$$

H is calculated according to equation (E):

$$H = \frac{L}{16\left(\frac{t}{w}\right)^2} \quad (E)$$

L was the length of the capillary column ($L=15$ m). w represents the full width of the analyte.” Please see Page S57, Section S15 in Supplementary Information.

Supplementary Figure 52. The Gouy plots of 2-methylpentane on (a) Zr-TCPE and (b) Zr-TCPE-H coated columns. The test temperature was 393.15 K. The kinetic diffusion constant (D_s) of 2-methylpentane on Zr-TCPE and Zr-TCPE-H coated column was $2.02 \times 10^{-12} \text{ m}^2/\text{s}$ and $6.71 \times 10^{-15} \text{ m}^2/\text{s}$, respectively. The calculated resistance to mass transfer coefficient (C_m) of 2-methylpentane on Zr-TCPE and Zr-TCPE-H coated column

was 7.10×10^{-4} and 0.25, respectively.

We have revised “Meanwhile, the longer heating time of Zr-TCPE led to a further decrease in resolution, indicating larger structural deformation” to “It was worth noting that, the separation performance of Zr-TCPE coated column decreased under the longer heating time, proving the structure change from Zr-TCPE to Zr-TCPE-*H*” Please see Page 7, right column, the first paragraph in the manuscript.

We have revised “This phenomenon proved the enhanced stability of Zr-TCPE-L₂ due to the rigidification of flexibility along the c-axis and the restriction of intramolecular ligand motion” to “This phenomenon proved the enhanced stability of Zr-TCPE-L₂. The coordination of L₂ at pocket B greatly rigidified the flexibility along the c-axis and restricted the intramolecular motion of organic ligands. Meanwhile, the L₂ dangled on pocket A restricted the decrease of pores along the a-axis. As a result, the pore size and pore orderliness of Zr-TCPE-L₂ were mostly maintained after heating, leading to enhanced separation stability” in the manuscript. Please see Page 7, right column, second paragraph in the manuscript.

We have revised “In addition, Zr-TCPE-L₁-*H* showed fair separation ability with obvious peak tailing compared with Zr-TCPE-L₁ (Supplementary Fig. 55 and 56), which is due to the existence of flexibility along the c-axis” to “However, Zr-TCPE-L₁-*H* showed fair separation ability with obvious peak tailing compared with Zr-TCPE-L₁ (Supplementary Fig. 55 and 56). Although the installation of L₁ at pocket A restricted the intermolecular motion of organic ligands on the ab plane, the intramolecular motion still existed. This intramolecular motion would result in the disordered orientation of phenyl rings in the pore channel, which led to reduced separation ability of Zr-TCPE-L₁-*H*” in the manuscript. Please see Page 7, right column, second paragraph in the manuscript.

We have added “Compared with Zr-TCPE, the separation resolution of nonane isomers on the Zr-TCPE-DLI coated column displayed almost no decrease with the extension of heating time, confirming the enhanced separation stability after linker

installation (Supplementary Fig. 58).” In the manuscript. Please see Page 8, left column, the first paragraph in the manuscript.

6. *The statement “It was worth noting that there was almost no separation efficiency loss in Zr-TCPE-DLI-H (Supplementary Fig. 43), indicating that not only its structural flexibility was completely rigidified” is in contrast to the pxrd data for this material (SI Fig. 23) where the Zr-TCPE-DLI-H phase shows a shift in the low angle peak towards lower 2theta values, suggesting an increase in the unit cell size and the presence of flexibility.*

Response: Thank you for pointing this out. We apologize for the mistake in the data processing. We have carefully rechecked the patterns and found that the PXRD pattern of Zr-TCPE-DLI-H was incorrectly assigned. We have revised the PXRD patterns in Supplementary Information. Please see Page S38, Supplementary Figure 30 in Supplementary Information.

Reviewer #3

Comments:

This work describes the rigidification of an scu topology ZrMOF via linker insertion. The scu topology has two distinct pocket and here two different linkers were installed on each pocket stepwise and the resulting materials were compared. The resulting MOFs were characterized extensively and deuterated analogue of final some MOFs were prepared to establish structural rigidity via solid state NMR. While the topology, the MOF itself and linker insertion for rigidification is not new, it is worth publishing the demonstration of linker rotation and also the separation performance of the MOFs. However the following major points need to be addressed before its publication.

Response: We highly appreciate the referee for the supportive comments and we have improved the manuscript significantly according to your following comments.

1. Firstly, while the authors cited the previously published TCPE MOFs with biphenyl analogs, the same ligand was also used for building scu topology Zr MOF. Authors need to cite the original work (Zirconium-based metal–organic framework gels for selective luminescence sensing RSC Adv., 2020,10, 44912-44919) and mention this in both article and SI where the synthesis of this material is mentioned.

Response: Thank you for your suggestion. We have added the reference in the synthesis of Zr-TCPE. Please see **Page 10, Reference 40** in the manuscript.

2. While authors cited other ligand insertion studies, they missed an important one for this research which shows that insertion of the same naphlene based ligand used here into an scu Zr MOF which increased its mechanical properties, more specifically bulk modulus (Robison et al. Chem. Mater. 2020, 32, 8, 3545–3552)

Response: Thank you for pointing this out. We have added the reference in the manuscript. Please see **Page 10, Reference 38** in the manuscript.

3. *Were the Authors able to occupy all the possible sites to install these linkers? This should be mentioned in the discussion.*

Response: Thank you for your suggestion. We have already used excessive secondary linkers to try to occupy all the vacant pockets with secondary linkers but failed. Ideally, if the vacant coordination pockets in Zr-TCPE are fully occupied with installed linkers, the ligand ratio of L₁: L₂: TCPE will be 0.5: 0.5: 1. In Zr-TCPE-L₁, the ratio of L₁ to TCPE was 0.5: 1, which was consisted with the theoretical value. Thus, we supposed that pocket A was fully occupied by L₁ in Zr-TCPE-L₁ as the possibility of L₁ at pocket B was ruled out due to the size mismatching. While, the ratio of L₂ to TCPE was 0.4:1, which was smaller than the theoretical ratio. We supposed that the steric hindrance of pocket B in the original MOF prevented full coordination at pocket B. When inserting two linkers into the Zr-TCPE, the situation can be more complex. On the one hand, the L₂ was small enough to coordinate at pocket A as the unstable dangling linker. Thus, the L₂ will competitively coordinate towards pocket A, leading to the unsaturated coordination of pocket A. On the other hand, the steric hindrance of pocket B still prevents full coordination. Thus, we can hardly occupy the Zr-TCPE-DLI with all the possible sites to install these linkers.

We have further increased the amount of L₁ and L₂ in the synthesis procedure, however, the ratio of L₁: L₂: TCPE was 0.5: 0.2: 1, and the pocket B was not fully occupied. Besides, we also tried to add two linkers simultaneously in the synthesis, however, the ratio of L₁: L₂: TCPE was 0.4: 0.4: 1, and neither pocket was fully occupied. Please see Page S31-32, Supplementary Figure 24-25 in Supplementary Information.

We have added that “Further increasing the amount of L₁ and L₂ in the synthesis of Zr-TCPE-DLI, the ratio of L₁: L₂: TCPE was 0.5: 0.2: 1. The pocket B was still not fully occupied. Besides, adding L₁ and L₂ simultaneously in the synthesis process, the ratio of L₁: L₂: TCPE was 0.4: 0.4: 1, indicating neither pocket was fully occupied.” Please see Page S32 in Supplementary Information.

Supplementary Figure 24. The ¹H NMR spectroscopy of digested Zr-TCPE-DLI with twice the amounts of linkers (TCPE: L₁: L₂: AA = 1: 0.5: 0.2: 0.06).

Supplementary Figure 25. The ¹H NMR spectroscopy of digested Zr-TCPE-DLI with the addition of L₁ and L₂ simultaneously (TCPE: L₁: L₂: AA = 1: 0.4: 0.4: 0.01).

4. Authors reported the BET surface area of the materials studied here with decimal points. As it was mentioned in a recent Tutorial article (DOI: <https://doi.org/10.1039/D1TA08021K>), the surface areas obtained from the BET equation is subject to error depending on the pressure region selected so it is not accurate to report surface areas with this precision as it is not that reproducible. So I recommend authors to either remove the decimal points or report the number with error bars.

Response: Thank you for your suggestion. We have revised “The Brunauer-Emmett-Teller (BET) surface area of Zr-TCPE was 934.27 m²/g” to “The Brunauer-Emmett-Teller (BET) surface area of Zr-TCPE was 934 m²/g.” Please see **Page 3, right column, second paragraph** in the manuscript. We have revised “The BET surface area of Zr-TCPE-L₁ and Zr-TCPE-L₂ were 670.63 m²/g and 894.14 m²/g, respectively” to “The BET surface area of Zr-TCPE-L₁ and Zr-TCPE-L₂ were 670 m²/g and 894 m²/g, respectively.” Please see **Page 5, right column, second paragraph** in the manuscript.

5. Authors mentioned that they were not able to get the single crystal despite many trials. Their experimental section shows that they add water as well in addition to acetic acid. Water typically results in smaller particles. It could be because of the fact that they are getting other phases such as csq, in that case I recommend not using water but using a little more DMF. Not sure if this would result in diffraction quality crystals but I am nearly confident that this would give that larger crystals.

Response: Thank you for your suggestion. We have tried the synthesis of Zr-TCPE without water but with more DMF, there were still no single crystals obtained. We have also tried other synthesis methods, and despite the increase in particle size, no diffraction-quality crystals were obtained.

We have carried out the trial synthesis of single crystals with more DMF: 10 mg ZrCl₄ and 120 μL AA were dissolved in 2/3/4 mL DMF in a glass vial. The vial was heated at 100 °C for 1 h. After cooling down to room temperature, 160 μL AA, and 10 mg H₄TCPE ligand

were added to the mixture. After sonication, the vial was heated at 120 °C for 24 h. The white product was collected by centrifugation and washed with DMF and EtOH three times, respectively, before drying at 60 °C under vacuum. The particle size of Zr-TCPE slightly increased with the increase of DMF content, but the size was non-uniform and these materials were not suitable for single crystal X-ray diffraction.

Figure R1. The SEM images of Zr-TCPE without the addition of H₂O, and with more DMF. (a,b) 2mL DMF, (c,d) 3 mL DMF, and (e,f) 4 mL DMF.

Figure R2. The PXRD patterns of Zr-TCPE without the addition of H₂O, but with more DMF.

We have carried out the trial synthesis of single crystals with BA as a modulator: The 10 mg ZrCl₄, 200/300/400/600/800 mg BA, and 10 mg H₄TCPE were dissolved in 2 mL DMF in a glass vial. The vial was heated at 120 °C for 24 h or 72 h. After cooling down to room temperature, the product was collected by centrifugation and washed with DMF and EtOH three times, respectively, before drying at 60 °C under vacuum. The particle size of Zr-TCPE-BA increased with the increase of BA content, but these materials were not suitable for single-crystal X-ray diffraction.

Figure R3. The SEM images of Zr-TCPE-BA with the addition of (a) 200 mg BA, (b) 300 mg BA, (c) 400 mg BA, (d) 400 mg BA (extending reaction time to 72 h), (e) 600 mg BA, and (f) 800 mg BA.

Figure R4. The PXRD patterns of Zr-TCPE-300 mg BA and Zr-TCPE-400 mg BA.

We have carried out the trial synthesis of single crystals with TFA as a modulator: The 10 mg $ZrCl_4$, 50/100/150/200 μL TFA, and 10 mg H_4TCPE were dissolved in 2 mL DMF in a glass vial. The vial was heated at 120 $^{\circ}C$ for 24 h. After cooling down to room temperature, the product was collected by centrifugation and washed with DMF and EtOH three times, respectively, before drying at 60 $^{\circ}C$ under vacuum. The particle size of Zr-TCPE-TFA increased with the increase in TFA content, but these materials were not suitable for single-crystal X-ray diffraction.

Figure R5. The SEM images of Zr-TCPE-TFA with the addition of (a) 50 μL TFA, (b) 100 μL TFA, (c) 150 μL TFA, and (d) 200 μL TFA.

Figure R6. The PXRD patterns of Zr-TCPE-200 μL TFA.

6. Authors also coated the inner surface of a column with the MOF to use it for hydrocarbon separation. However, their experimental section is not clear about exactly how the procedure was one. They mentioned it was x mL suspension was passed through column but they did not mention how this was done. This is an important step given that the capillary diameter is so small. I recommend authors to add more details to their experimental section.

Response: Thank you for pointing this out. We have revised the “1 mL methanol suspension of each material (2 mg/mL) was first filled into the capillary column and then pushed through the column at a velocity of 4.5 mL/min to leave a wet coating layer on the inner wall of the capillary column.” to “Typicall, 1 mL methanol suspension of each material (2 mg/mL) was first filled into the insulin syringe. The insulin syringe was connected to the capillary column. Then the MOF suspension was pushed through the column by a syringe pump at a velocity of 4.5 mL/min to leave a wet coating layer on the inner wall of the capillary column.” Please see Page S58 in Supplementary Information. We have added the schematic illustration of coating MOF suspension into the capillary column. Please see Page S58, Supplementary Figure 54 in Supplementary Information.

Supplementary Figure 54. The illustration of coating materials into the capillary column.

7. Similar to above, for linker insertion, authors have cited a previous work without mentioning the amount of MOF used for each reaction. The reagents amount needs to be

added to ensure others can safely reproduce your results.

Response: Thank you for pointing this out. We have revised “Typically, as-synthesized Zr-TCPE was soaked in DMF (2 mL) with the addition of the secondary linkers (0.03 M).” to “Typically, as-synthesized Zr-TCPE (about 15 mg) was soaked in DMF (2 mL) with the addition of the secondary linkers (0.03 M, 4.6 mL DMF).” Please see **Page 9, left column, second paragraph** in the manuscript. We have revised “Typically, as-synthesized Zr-TCPE was soaked in DMF (2 mL) with the addition of linker L₂ (0.03 M, 4.6 mL DMF).” to “Typically, as-synthesized Zr-TCPE (about 15 mg) was soaked in DMF (2 mL) with the addition of linker L₂ (0.03 M, 4.6 mL DMF).” Please see **Page S5** in Supplementary Information.

Reviewers' Comments:

Reviewer #1:

Remarks to the Author:

It is always gratifying when an author chooses to take on board referees' comments, and uses them to improve the quality of the paper. The clarification of the experimental details makes this work more scientifically sound and significantly enhance the importance of this paper, and I am now happy to recommend acceptance.

Reviewer #2:

Remarks to the Author:

Dear Editor,

The revised manuscript is improved. However, the quality of the experimental data, the discussion and clarity of the results, as well as the overall originality and novelty of the work, are not suitable for publication to Nature Communications. I recommend publication in a more specialized journal after taking into account the following comments.

1. The assignment to a non-interpenetrated scu structure is not convincing. This is mainly for the following reasons:

a) Pawley (and not Powley as written in the legend of Fig. 1) refinement is a structureless methodology. This means that there are no atoms and atomic positions involved in the refinement, as in the case of Rietveld refinement. Therefore, the question is: how the authors obtained the structural images shown in the manuscript and in the SI file? The same question holds for the structure file used for the theoretical calculation. Did the authors apply a Rietveld refinement in their experimental pxrd data? If yes they should provide all the corresponding details, including where they obtained the initial structural model.

b) It is evident from the HRTEM images that there is a superposition of single crystals. For accurate assignment of lattice distances HRTEM images from single crystals are necessary.

2. Perhaps a direct method for the assignment of a non-interpenetrated scu structure is to record accurate gas sorption isotherms such as N₂ at 77K but preferable Ar at 87 K (accurate micropore analysis starting from very low relative pressures, eg 10⁻⁶ p/p₀), from which accurate pore size distribution data can be extracted. A non-interpenetrated scu is expected to have a significant larger pore size and also a significantly higher total pore volume compared to a interpenetrated scu. This kind of data is not available and the authors are strongly advised to execute these experiments.

Reviewer #3:

Remarks to the Author:

Authors have addressed all my comments and other reviewers comments. To me they also were successful on defending the novelty of their work regarding the comments to second reviewer. Therefore I recommend current work for publication.

Response to Reviewers' Comments

Reviewer: #1

Comments:

It is always gratifying when an author chooses to take on board referees' comments, and uses them to improve the quality of the paper. The clarification of the experimental details makes this work more scientifically sound and significantly enhance the importance of this paper, and I am now happy to recommend acceptance.

Response: We highly appreciate your supportive comments.

Response to Reviewers' Comments

Reviewer: #2

Comments:

The revised manuscript is improved. However, the quality of the experimental data, the discussion and clarity of the results, as well as the overall originality and novelty of the work, are not suitable for publication to Nature Communications. I recommend publication in a more specialized journal after taking into account the following comments.

Response: We highly appreciate your constructive comments. We totally understand your concern about the non-interpenetration of scu structure from the technical aspect. We have added Rietveld refinement, ultrahigh-resolution low-dose HRTEM measurement, and Ar adsorption-desorption measurement at 87 K to confidently confirm the successful synthesis of non-interpenetration of scu structure.

Furthermore, we would like to re-emphasize the originality and novelty of this work. The scientific concepts, anisotropic flexibility of a TPE-based Zr-MOF, and its anisotropic rigidification are first proposed in this manuscript. **First**, the synthesis of a non-interpenetrated TPE-based Zr-MOF is a prerequisite. We have independently synthesized the non-interpenetrated Zr-TCPE, which is coincidentally similar to a recent publication (Ref#29, *J. Am. Chem. Soc.* 2023, 145, 1072-1082). The statement of independence is clearly supported by many experimental details, such as the synthesis methods (two-step versus one-pot) and structural characterizations (combination of XRD and HAADF versus Rietveld refinement). **Second**, the anisotropic stimuli-responsive emission property of Zr-TCPE was then discovered under different temperatures. The maximum emission wavelength exhibited apparent red-shift PL properties. We contributed this PL change to the inter- and intra-molecular motion of organic ligands and performed the linker installation strategy to anisotropically control each motion. The mechanism and modulation method for PL changes is different from the previous reports. **Third**, we have also employed numerous characterization methods to support the non-interpenetration structure as well as the anisotropic flexibility and rigidification, such as ultrahigh-resolution low-dose HRTEM

imaging, HAADF images, XRD patterns, Raman spectra, *in-situ* ^2H solid-state NMR, DFT calculations, rational design of installed linkers, fluorescence emission spectra, stability test, and improved GC separation performance. Some techniques were first employed to reveal anisotropic flexibility and rigidification of MOFs. **Thus**, from the viewpoint of anisotropic flexibility and its rigidification, the re-evaluation of the originality and novelty of this manuscript is suggested.

1. The assignment to a non-interpenetrated scu structure is not convincing. This is mainly for the following reasons:

a) Pawley (and not Powley as written in the legend of Fig. 1) refinement is a structureless methodology. This means that there are no atoms and atomic positions involved in the refinement, as in the case of Rietveld refinement. Therefore, the question is: how the authors obtained the structural images shown in the manuscript and in the SI file? The same question holds for the structure file used for the theoretical calculation. Did the authors apply a Rietveld refinement in their experimental pxrd data? If yes they should provide all the corresponding details, including where they obtained the initial structural model.

Response 1(a): Thank you for pointing this out. We have added the details about Rietveld refinement of Zr-TCPE to show more accurate structural information, including where we obtained the initial structural model. The refinement confirms the non-interpenetrated **scu** topology of Zr-TCPE. The refinement reveals that Zr-TCPE crystallizes in orthorhombic crystal system with *Cmmm* space group. Other crystallographic parameters, such as a, b, and c axis parameters are given in Supplementary Table 1. The atoms and atomic positions in Zr-TCPE are listed in Supplementary Table 2. Besides, the diffraction parameters of Zr-TCPE, including d-spacing of diffraction planes are given in Supplementary Table 3. **Please see Page S6-9, Supplementary Figure 1, and Table 1-3 in Supplementary Information.**

We have revised “Powley” to “Pawley”.

We have revised “Thus, the powder X-ray diffraction (PXR) refinement was utilized to reveal the structure of this material” to “Thus, the powder X-ray diffraction

(PXRD) Rietveld refinement was utilized to reveal the structure of this material (Supplementary Fig. 1)". Please see Page 2, left column, last paragraph in the manuscript.

We have revised "Detailed crystallographic parameters were given in Supplementary Information (Supplementary Table 1)" to "Detailed crystallographic parameters, atomic positions, and diffraction parameters were given in Supplementary Information (Supplementary Table 1-3)". Please see Page 2, right column, last paragraph in the manuscript.

We have added "**Rietveld Refinement:** High-quality PXRD data for Rietveld refinement was collected on Rigaku SmartLab 9 Kw (Tokyo, Japan) diffractometer with a $\text{CuK}\alpha$ radiation (1.54056 Å, angle range: 4-110°, step size: 0.01°, IS=1/4, RS1=5 mm, room temperature). All the processes of refinements were performed on the TOPAS 64 V6" in Supplementary Information. Please see Page S4 in Supplementary Information.

We have added "**Refinement details of Zr-TCPE:** A structural model of Zr-TCPE was developed starting from NU-901 structure.² Indexing of the PXRD data of Zr-TCPE suggested orthorhombic space groups. Assuming the inorganic building unit to be the hexanuclear cluster most frequently observed in Zr-MOFs, a structure model starting from the NU-901 was set up. The first step was replacing the ligand from TBAPy in NU-901 to TCPE and adjusting the original lattice parameters to the ones obtained by indexing. Molecular mechanics optimizations of the atom positions within the unit cell were performed to reach energy minimization using the Universal force field (UFF) in Material Studio 7.0. Interpenetration was not possible in this case due to limited lattice void space. Pawley refinement (Reflex module of Materials Studio) against the full powder pattern using profile fitting, FWHM, and asymmetry correction parameters yields cell parameters $a = 17.724(46)$ Å, $b = 30.182(02)$ Å, and $c = 12.292(77)$ Å (residuals: $R_p = 4.40\%$, $R_{wp} = 6.41\%$). The low residuals indicated the refined profile matched the experimental XRD pattern very well. To examine the model, Rietveld refinement was also performed on the TOPAS 64 V6." in Supplementary Information. Please see Page S6 in Supplementary Information.

We have added the reference:

Robison, L. et al. Designing porous materials to resist compression: mechanical reinforcement of a Zr-MOF with structural linkers. *Chem. Mater.* 32, 3545-3552 (2020).

Please see Page S75 in Supplementary Information.

Supplementary Figure 1. Final Rietveld refinement plots of Zr-TCPE. The experimental, calculated, and difference curves are in blue, red, and gray, respectively. The vertical bars indicate the positions of the Bragg peaks.

Supplementary Table 1. Crystallographic parameters of synthesized Zr-TCPE.

Compound	Zr-TCPE
Crystal system	Orthorhombic
Space group	Cmmm
Radiation	CuK α ($\lambda=1.54056$ Å)
a/Å	17.5111
b/Å	30.1849
c/Å	12.1237
V/Å ³	6408.20
α	90.00000
β	90.00000
γ	90.00000
R _{wp} /%	13.81
R _p /%	10.68
R _{Bragg} /%	2.77
GOF	3.98

Supplementary Table 2. Atomic parameters of Zr-TCPE.

Atom	x/a	y/b	z/c
O1	0.56037	0.61971	0.79012
O2	0.67488	0.61874	0.5577
C3	0.68413	0.66037	0.80988
C4	0.73336	0.70335	0.83599
C5	0.75656	0.69527	0.83985
H6	0.75226	0.64777	0.95985
C7	0.755	0.69553	0.83847
H8	0.72529	0.6694	0.79736
C9	0.59055	0.72943	0.51884
C10	0.66646	0.7544	0.90941
C11	0.62371	0.67245	0.83232
H12	0.40513	0.73503	1.45902
H13	0.22274	0.83402	1.20438
Zr14	0.38698	0.48315	0.36164
O15	0.45017	0.50124	0.21789
O16	0.26593	0.48604	0.29215
Zr17	0.46672	0.57749	0.44966
O18	0.50335	0.53726	0.32087
O19	0.27924	0.56993	0.56334
C20	0.68962	0.72665	0.85677

Supplementary Table 3. Main diffraction parameters of Zr-TCPE.

(hkl)	2 Theta (degree)	d-spacing (nm)
(020)	5.85	1.51
(001)	7.29	1.21
(021)	9.35	0.94
(130)	10.13	0.87
(040)	11.72	0.75
(131)	12.49	0.71

b) It is evident from the HRTEM images that there is a superposition of single crystals. For accurate assignment of lattice distances HRTEM images from single crystals are necessary.

Response 1(b): Thank you for your constructive suggestion. To overcome the superposition issue in HRTEM, we have collaborated with Prof. Yu Han at KAUST, a leading expert who devised the methodology for the procurement of atomically resolved images of beam-sensitive materials, such as MOFs using TEM. We have then added the low-dose (only a few electrons per square angstrom) HRTEM images of Zr-TCPE single crystals with enhanced resolution to successfully confirm the non-interpenetration structure. **Please see Page 3, Figure 2 in the manuscript, and Page S12-14, Supplementary Figure 3-8 in Supplementary Information.**

We have added “The non-interpenetrated structure of Zr-TCPE was further confirmed by low-dose high-resolution transmission electron microscopy (HRTEM) imaging (Fig. 1b).⁴¹⁻⁴³ The low electron dose (only a few electrons per square angstrom) of HRTEM avoids the structural damage of Zr-TCPE under electron beams. The black dots in the HRTEM images represented the Zr₆ clusters. The fast Fourier transform (FFT) pattern of the marked area in the HRTEM image was acquired. The calculated d-spacing of (020), (001), and (021) was 1.53 nm, 1.26nm, and 0.95 nm, respectively, which was consistent with the d-spacing from PXRD refinement (Fig. 1c and Supplementary Table 3). For better interpretation, the raw image was processed by

correcting the effect of the contrast transfer function (CTF) of the objective lens (Supplementary Fig. 5). Furthermore, the simulated electron diffraction (ED) pattern along the [100] direction was consistent with the selected area electron diffraction (SAED) pattern of the HRTEM image (Supplementary Fig. 6). The average background subtraction filter (ABSF)-filtered CTF-corrected image in Fig. 1d matched well with the simulated structure of Zr-TCPE along the [100] direction zone axis (Supplementary Fig. 3). The distances between two adjacent Zr₆ clusters were measured from the linear profiling along the b- and c-axis as 1.53 nm and 1.23 nm, respectively (Supplementary Fig. 7), which were consistent with the simulated structure of Zr-TCPE (Supplementary Fig. 3). Besides, the distance of Zr₆ clusters along the c-axis was comparable to the c-axis cell parameter in non-interpenetrated Zr-TCPE. Thus, not interpenetrated but non-interpenetrated structure was assigned to Zr-TCPE. We also obtained diffraction information of (020) and (130) planes from the FFT images (Supplementary Fig. 8). All the above results verified the non-interpenetrated structure of Zr-TCPE.” in the manuscript. Please see Page 3, left column, first paragraph in the manuscript. Please see Page S12-14, Supplementary Figure 5-8 in Supplementary Information.

We have added the following references:

1. Guo, F.-A. et al. Linker vacancy engineering of a robust ftw-type Zr-MOF for hexane isomers separation. *Angew. Chem. Int. Ed.* 62, e202303527 (2023).
2. Liu, L. et al. Imaging defects and their evolution in a metal-organic framework at sub-unit-cell resolution. *Nat. Chem.* 11, 622-628 (2019).
3. Liu, G. et al. Eliminating lattice defects in metal-organic framework molecular-sieving membranes. *Nat. Mater.* 22, 769-776 (2023). Please see Page 10, right column in the manuscript.

We have added “**Low-dose HRTEM measurements:** low-dose HRTEM experiments were performed on a Cs-corrected FEI cubed G2 Titan 60-300 electron microscope at an acceleration voltage of 300 kV, using a Gatan K2 direct-detection camera in the electron-counting mode with the dose fractionation function. The ED pattern was simulated by using Single Crystal software” in Supplementary Information. Please see Page S4 in Supplementary Information.

We have added “On the one hand, the structure of Zr-TCPE was analyzed by Rietveld refinement. The refinement results revealed that each TCPE ligand was connected to four Zr_6 clusters and each Zr_6 cluster was coordinated with eight TCPE ligands. Then, the non-interpenetrated **scu** coordination structure was generated in Zr-TCPE. On the other hand, the low-dose HRTEM image acquired along the [100] direction further confirmed the non-interpenetrated structure. The arrangement of Zr_6 clusters on the ABSF-filtered CTF-corrected image was consistent with the simulated non-interpenetrated structure of Zr-TCPE along the [100] direction. The calculated distances of adjacent Zr_6 clusters were also comparable to the simulated values in non-interpenetrated Zr-TCPE along the [100] direction. Besides, the calculated d-spacing of (020), (001), and (021) planes from the FFT pattern was similar to the simulated value. All the results revealed the non-interpenetrated structure of Zr-TCPE.” in Supplementary Information. Please see Page S13 in Supplementary Information.

We have added “We can reasonably speculate that if Zr-TCPE possesses the interpenetrated structure, the distance of adjacent Zr_6 clusters along the c-axis will be half of the c-axis cell parameter (Supplementary Fig. 4). This hypothesis is inconsistent with the actual experimental value in HRTEM images. The distance of adjacent Zr_6 clusters along the c-axis calculated from the HRTEM image is 1.25 nm, which is consistent with the value in simulated non-interpenetrated Zr-TCPE (1.21 nm). Thus, non-interpenetrated structure not interpenetrated one is assigned to Zr-TCPE” in Supplementary Information. Please see Page S13 in Supplementary Information.

Fig.1 Schematic illustration and characterization of Zr-TCPE. **a**, The construction of Zr-TCPE from Zr_6 cluster and H_4TCPE ligand. The illustration of pocket A along [001] direction and pocket B along [100] direction in Zr-TCPE. **b**, HRTEM image of Zr-TCPE acquired along the [100] direction. **c**, FFT pattern of the marked region in HRTEM image. The d-spacing of (020), (021), and (001) were calculated at 1.53 nm, 0.95 nm, and 1.23 nm, respectively. **d**, ABSF-filtered CTF-corrected image of the marked region in HRTEM image. The black dots in the HRTEM images represented the Zr_6 clusters. **e**, The locally enlarged image. The TCPE ligand was even observed in the center of four adjacent Zr_6 clusters. **f**, Simulated structure of Zr-TCPE along [100] direction.

Supplementary Figure 3. The simulated structure of non-interpenetrated Zr-TCPE along (a) the [001] direction and (b) the [100] direction.

Supplementary Figure 4. The simulated structure of interpenetrated Zr-TCPE along (a) the [001] direction and (b) the [100] direction.

Supplementary Figure 5. (a) The raw low-dose HRTEM image of Zr-TCPE. The black dots in the HRTEM images represented the Zr_6 clusters. (b) CFT-corrected image. (c) ABSF-filtered CTF-corrected image.

Supplementary Figure 6. (a) The low-dose HRTEM image of Zr-TCPE along the [100] direction. The black dots in the HRTEM images represented the Zr_6 clusters. The insert in the FFT pattern of the marked region. (b) The SAED pattern of Zr-TCPE. (c) Simulated electron diffraction pattern along the [100] direction.

Supplementary Figure 7. (a) The raw low-dose HRTEM image of Zr-TCPE along the [100] direction. The black dots in the HRTEM images represented the Zr_6 clusters. (b,c) The measured distance of adjacent Zr_6 clusters was along the b-axis (labeled in blue) and the c-axis (labeled in yellow). The distance of adjacent Zr_6 clusters along the b-axis and c-axis was around 1.53 nm and 1.23 nm, which was comparable to the simulated

value (1.51 nm and 1.21 nm shown in Supplementary Fig.3).

Supplementary Figure 8. The HRTEM image of Zr-TCPE with FFT patterns of marked areas. The calculated d-spacing of (020) and (130) planes were consistent with the simulated values ($d_{(020)}=1.51$ nm, $d_{(130)}=0.87$ nm).

2. Perhaps a direct method for the assignment of a non-interpenetrated scu structure is to record accurate gas sorption isotherms such as N_2 at 77K but preferable Ar at 87 K (accurate micropore analysis starting from very low relative pressures, eg 10^{-6} p/p_0), from which accurate pore size distribution data can be extracted. A non-interpenetrated scu is expected to have a significant larger pore size and also a significantly higher total pore volume compared to a interpenetrated scu. This kind of data is not available and the authors are strongly advised to execute these experiments.

Response 2: Thank you for pointing this out. To confirm the non-interpenetration structure, we have added the Ar adsorption isotherms of Zr-TCPE at 87K starting from very low relative pressure (3.17×10^{-6} P/P_0). The BET surface area, pore size distribution, and total pore volume obtained from the Ar adsorption isotherms are consistent with those from N_2 adsorption isotherms. The main pore size distribution of Zr-TCPE calculated from the Ar adsorption isotherms was 8.7 Å, which was slightly smaller than

the value (9.4 Å) obtained from N₂ adsorption. This measured pore size was consistent with the simulated value (9.3 Å) of pocket A in Zr-TCPE. This result indicated the non-interpenetrated structure of Zr-TCPE. **Please see Page S19, Supplementary Figure 13, and Supplementary Table 4 in Supplementary Information.**

We have revised “To characterize the porosity of Zr-TCPE, N₂ adsorption-desorption isotherms were performed at 77 K. As shown in Supplementary Fig. 8, the isotherms presented fully reversible type-I behavior, indicating the microporous characteristic of Zr-TCPE. The Brunauer-Emmett-Teller (BET) surface area of Zr-TCPE was 934 m²/g and the pore size distribution of Zr-TCPE calculated by the DFT method was 9.4 Å, which was consistent with the simulated size of pocket A” to “To accurately characterize the porosity of Zr-TCPE, both the N₂ and Ar adsorption-desorption isotherms were performed at 77 K and 87 K, respectively. As shown in **Supplementary Fig. 13**, both the isotherms presented fully reversible type-I behavior, indicating the microporous characteristic of Zr-TCPE. The Brunauer-Emmett-Teller (BET) surface area, pore size distribution, and total pore volume obtained from the Ar adsorption isotherms are consistent with those from N₂ adsorption isotherms (**Supplementary Table 4**). The main pore size of Zr-TCPE calculated by the DFT method from Ar and N₂ sorption isotherms was 8.7 Å and 9.4 Å, respectively. The calculated pore size was comparable to the simulated size of pocket A in Zr-TCPE (9.3 Å), indicating the non-interpenetrated feature of Zr-TCPE” In the manuscript. **Please see Page 4, left column, first paragraph in the manuscript.**

Supplementary Figure 13. (a) The N₂ adsorption-desorption isotherms and (b) pore size distribution of Zr-TCPE measured at 77 K. (c) The Ar adsorption-desorption isotherms and (d) pore size distribution of Zr-TCPE measured at 87K.

Supplementary Table 4. Porosity parameters of Zr-TCPE from N₂ and Ar adsorption isotherms.

Adsorbate	BET surface area (m ² /g)	Main pore size (Å)	Total pore volume (cm ³ /g)
N ₂	934	9.4	0.67
Ar	955	8.7	0.64

Response to Reviewers' Comments

Reviewer: #3

Comments:

Authors have addressed all my comments and other reviewers comments. To me they also were successful on defending the novelty of their work regarding the comments to second reviewer. Therefore I recommend current work for publication.

Response: We highly appreciate your supportive comments.

Reviewers' Comments:

Reviewer #2:

Remarks to the Author:

[Note from the editor: Reviewer #2 made comments to the Editor only and confirms that the technical concerns have been addressed.]